

# Gaseous, PM$_{2.5}$ mass, and speciated emission factors from laboratory chamber peat combustion

**John G. Watson**[1,2], **Junji Cao**[2,3], **L.-W. Antony Chen**[4], **Qiyuan Wang**[2], **Jie Tian**[2,3], **Xiaoliang Wang**[1], **Steven Gronstal**[1], **Steven Sai Hang Ho**[5], **Adam C. Watts**[1], and **Judith C. Chow**[1,2]

[1]Division of Atmospheric Sciences, Desert Research Institute, Reno, Nevada, USA
[2]Key Laboratory of Aerosol Chemistry and Physics, Institute of Earth Environment,
Chinese Academy of Sciences, Xi'an, China
[3]CAS Center for Excellence in Quaternary Science and Global Change, Chinese Academy of Sciences, Xi'an, China
[4]Department of Environmental and Occupational Health, University of Nevada, Las Vegas, Nevada, USA
[5]Hong Kong Premium Services and Research Laboratory, Hong Kong, China

**Correspondence:** John G. Watson (john.watson@dri.edu)

**Abstract.** TS1 Peat fuels representing four biomes of boreal (western Russia and Siberia), temperate (northern Alaska, USA), subtropical (northern and southern Florida, USA), and tropical (Borneo, Malaysia) regions were burned in a laboratory chamber to determine gas and particle emission factors (EFs). Tests with 25 % fuel moisture were conducted with predominant smoldering combustion conditions (average modified combustion efficiency (MCE) $= 0.82 \pm 0.08$). Average fuel-based EF$_{CO_2}$ (carbon dioxide) are highest $(1400 \pm 38\,\mathrm{g\,kg^{-1}})$ and lowest $(1073 \pm 63\,\mathrm{g\,kg^{-1}})$ for the Alaskan and Russian peats, respectively. EF$_{CO}$ (carbon monoxide) and EF$_{CH_4}$ (methane) are $\sim 12\,\%$–$15\,\%$ and $\sim 0.3\,\%$–$0.9\,\%$ of EF$_{CO_2}$, in the range of 157–171 and 3–10 g kg$^{-1}$, respectively. EFs for nitrogen species are at the same magnitude as EF$_{CH_4}$, with an average of $5.6 \pm 4.8$ and $4.7 \pm 3.1\,\mathrm{g\,kg^{-1}}$ for EF$_{NH_3}$ (ammonia) and EF$_{HCN}$ (hydrogen cyanide); $1.9 \pm 1.1\,\mathrm{g\,kg^{-1}}$ for EF$_{NO_x}$ (nitrogen oxides); and $2.4 \pm 1.4$ and $2.0 \pm 0.7\,\mathrm{g\,kg^{-1}}$ for EF$_{NO_y}$ (total reactive nitrogen) and EF$_{N_2O}$ (nitrous oxide).

An oxidation flow reactor (OFR) was used to simulate atmospheric aging times of $\sim 2$ and $\sim 7$ d to compare fresh (upstream) and aged (downstream) emissions. Filter-based EF$_{PM_{2.5}}$ varied by $> 4$-fold (14–61 g kg$^{-1}$) without appreciable changes between fresh and aged emissions. The majority of EF$_{PM_{2.5}}$ consists of EF$_{OC}$ (organic carbon), with EF$_{OC}$ / EF$_{PM_{2.5}}$ ratios in the range of 52 %–98 % for fresh emissions and $\sim 15$ % degradation after aging. Reductions of EF$_{OC}$ ($\sim 7$–$9\,\mathrm{g\,kg^{-1}}$) after aging are most apparent for boreal peats, with the largest degradation in low-temperature OC1 that evolves at $< 140\,^\circ$C, indicating the loss of high-vapor-pressure semivolatile organic compounds upon aging. The highest EF$_{Levoglucosan}$ is found for Russian peat $(\sim 16\,\mathrm{g\,kg^{-1}})$, with $\sim 35\,\%$–$50\,\%$ degradation after aging. EFs for water-soluble OC (EF$_{WSOC}$) account for $\sim 20\,\%$–$62\,\%$ of fresh EF$_{OC}$.

The majority ($> 95\,\%$) of the total emitted carbon is in the gas phase, with 54 %–75 % CO$_2$, followed by 8 %–30 % CO. Nitrogen in the measured species explains 24 %–52 % of the consumed fuel nitrogen, with an average of $35 \pm 11\,\%$, consistent with past studies that report $\sim 1/3$ to 2/3 of the fuel nitrogen measured in biomass smoke. The majority ($> 99\,\%$) of the total emitted nitrogen is in the gas phase, with an average of 16.7 % as NH$_3$ and 9.5 % as HCN. N$_2$O and NO$_y$ constituted 5.7 % and 2.9 % of consumed fuel nitrogen. EFs from this study can be used to refine current emission inventories.

## 1 Introduction

Globally, peatlands occupy $\sim 3\,\%$ of the Earth's land surface, but they store as much as 610 gigatonnes (i.e., $610 \times 10^{15}$ g) of carbon, representing 20 %–30 % of the planet's terrestrial carbon (Page et al., 2011; Rein et al., 2009). Peatland fires

can persist for weeks to months and are dominated by the smoldering phase as opposed to the flaming phase of biomass burning (Stockwell et al., 2016; Hu et al., 2018). This results in lower combustion efficiencies, increased particulate matter (PM) emissions, and larger fractions of brown carbon (BrC) compared to black carbon (BC) or soot (Pokhrel et al., 2016). Peat fires emit reduced nitrogen compounds (e.g., ammonia, NH$_3$; and hydrogen cyanide, HCN); volatile and semivolatile organic compounds (VOCs and SVOCs); and PM$_{2.5}$ (PM with aerodynamic diameters < 2.5 µm) (Akagi et al., 2011; Yokelson et al., 2013). Peat smoke and ash affect ecosystem productivity, soil acidity, biogeochemical cycling, atmospheric chemistry, Earth's radiation balance, and human health. Warmer climates lower the water table in peatlands and change the pattern, frequency, and intensity of the peat-land fires causing local- and regional-scale air pollution and visibility impairment (Page et al., 2002; Turetsky et al., 2010, 2015a, b). For Southeast Asia, fire-related regional air pollution and its effects on atmospheric visibility, ecosystems, and human health have been addressed in many studies (e.g., Behera et al., 2015; Betha et al., 2013; Bin Abas et al., 2004; Engling et al., 2014; Heil and Goldammer, 2001; Kundu et al., 2010; Levine, 1999; Hu et al., 2019; Tham et al., 2019; Fujii et al., 2017; Dall'Osto et al., 2014).

Nitrogen, one of the most important plant nutrients, affects global carbon and biogeochemical cycles (Crutzen and Andreae, 1990; Gruber and Galloway, 2008). Deposition of oxidized and reduced nitrogen species from biomass burning, such as gaseous nitric oxide (NO), nitrogen dioxide (NO$_2$), and NH$_3$ as well as particulate nitrate (NO$_3^-$) and ammonium (NH$_4^+$), alters terrestrial ecosystems (Chen et al., 2010), while nitric acid (HNO$_3$) contributes to soil acidification and excessive nitrification that reduce plant resistance to environmental stresses (Goulding et al., 1998). Gaseous nitrogen oxides (NO$_x$) affect atmospheric chemistry through (1) reactions with hydroxyl (OH) and peroxy (HO$_2$ + RO$_2$) radicals; (2) conversion to nitrate radical (NO$_3$), dinitrogen pentoxide (N$_2$O$_5$), and acyl peroxy nitrates (particularly per-oxyacetyl nitrate, PAN), which are important NO$_x$ reservoirs; and (3) formation of ozone (O$_3$) and secondary organic aerosols (SOA) (Alvarado et al., 2010; Cubison et al., 2011; Ng et al., 2007). While NH$_3$ neutralizes HNO$_3$ to form particulate ammonium nitrate (NH$_4$NO$_3$), it may also react with alkanoic acids to form alkyl amides, nitriles, and ammonium salts that can also contribute to SOA formation (Na et al., 2007; Simoneit et al., 2003; Zhao et al., 2013). In addition, NH$_3$ interacts with SOA to form BrC that further influence the aerosol radiative forcing (Updyke et al., 2012).

This study quantifies peat burning emission factors (EFs) for fresh and aged multipollutant mixtures through controlled burns in a laboratory combustion chamber with atmospheric aging simulated by an oxidation flow reactor (OFR). These tests are applied to peat samples from diverse parts of the world.

## 2 Experiment

### 2.1 Fuel types

Peatlands are found all over the world, as illustrated in Fig. 1 (based on Yu et al., 2010), with large deposits found in the northern USA and Canada, northern Europe, Russia/Siberia, and southeast Asia. Eight types of peat fuels from different regions and climates were collected for testing, including boreal (i.e., Odintsovo, Russia; and Pskov, Siberia), temperate (i.e., black spruce forest, northern Alaska, USA), subtropical (i.e., northern (Putnam County Lakebed) CE1 and southern (Everglades National Park) Florida, USA; and Caohai and Gaopo, Guizhou, southwest China), and tropical (i.e., Borneo, Malaysia) peats.

Representative peat samples of 250–1150 g from the upper 20 cm of the peatland surface were excavated for each region indicated in Fig. 1. As peat is a heterogeneous mixture of decomposed plant material, it can be formed in different wetlands under changing climates and nutrient contents (Turetsky et al., 2015a). Supplement TS2 Fig. S1 shows that the appearance of peat fuels varies by region.

To quantify carbon (C), hydrogen (H), nitrogen (N), sulfur (S), and oxygen (O) content, ∼ 2–3 g of each peat fuel was dried in a vacuum oven (∼ 105 °C) for 2 h prior to elemental analysis (Thermo Flash-EA 1112 CHNS/O Analyzer, Waltham, MA, USA).

Import and export regulations (USDA, 2010) require high-temperature heating of soil/peat fuels as part of the sterilization process. Peat fuels were heated to 90 °C and weighed every 24 h to achieve a stable dry mass with ∼ 0.16 % moisture by weight content (after ∼ 96 h of heating). The low heating temperature (i.e., below the water boiling point) minimized VOC losses, although some compounds with high volatilities could have been removed at 90 °C. To better simulate the field conditions during peat fires, distilled–deionized water (DDW) was added to rehydrate the dry peat and achieve a fuel moisture of ∼ 25 % (by weight) before each experiment (Yatavelli et al., 2017). To examine the effects of fuel moisture on emissions, additional experiments ($n = 3$) were conducted at 60 % fuel moisture content (by weight) for the Putnam (FL) peat.

### 2.2 Experimental setup

The laboratory setup shown in Fig. 2 used a biomass combustion chamber with a volume of ∼ 8 m$^3$ (1.8 m ($W$) × 1.8 m ($L$) × 2.2 m ($H$)) (Tian et al., 2015). Instrument specifications and operating principles are shown in Table S1 in the Supplement. The chamber is made of 3 mm thick aluminum to withstand high-temperature heating. A blower supplied air filtered by a charcoal bed and a high-efficiency particulate air (HEPA) filter near the bottom of the chamber to remove background gas and particle contaminants. The ventilation rate was controlled by the blower and exhaust fan

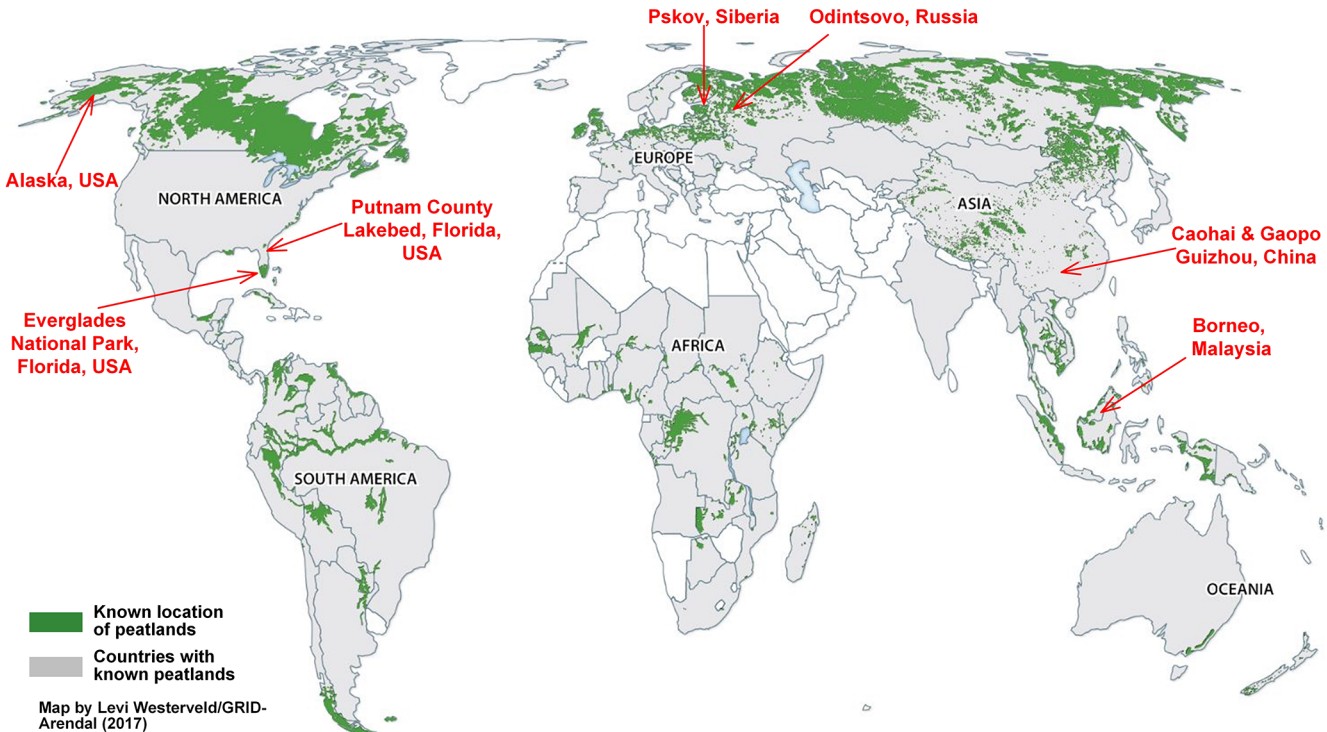

**Figure 1.** Global distribution of peatlands (based on Yu et al., 2010). Samples were obtained from Odintsovo, Russia; Pskov, Siberia; black spruce forest, northern Alaska, USA; Putnam County Lakebed and Everglades National Park, Florida, USA; Caohai and Gaopo, Guizhou, China; and Borneo, Malaysia.

at ∼ 2.65 m$^3$ min$^{-1}$, resulting in a smoke residence time of ∼ 3 min in the chamber assuming a well-stirred flow model.

For each test, ∼ 10–30 g of dried peat was placed in an asbestos-insulated circular container on top of an induction heater that provided heating during the first ∼ 5–10 min of combustion (see Fig. S2). This method replaced a propane torch used in initial test burns, thereby minimizing non-peat burning emissions. The smoldering process is usually self-propagating and sustained by heat conduction and radiation, with fuel mass continuously monitored by a scale underneath the induction heater (Ohlemiller et al., 1979).

Continuous PM$_{2.5}$ mass concentrations were monitored with a DustTrak (TSI model 8532, Shoreview, MN, USA) (Wang et al., 2009) (Table S1). When PM$_{2.5}$ concentrations reached their maximum and started to decline, the induction heater was turned off. The fuel was consumed with diminished smoke emissions after ∼ 20 min. Preliminary tests were conducted using ∼ 10–20 g of fuel and a dilution ratio of ∼ 3 to 5, yielding sufficient particle loadings on the filters (∼ 150–290 µg per 47 mm filter disc). To achieve higher filter deposits of 300–600 µg per filter that accommodate comprehensive organic speciation, additional fuels (∼ 15–20 g) were added with the induction heater turned on for another ∼ 10 min. Sampling continued until the concentrations returned to background level.

Sampling ports for stack concentrations of carbon dioxide (CO$_2$) and multiple gases by Fourier transform infrared (FTIR; model DX 4015; Gasmet Technologies Oy, Finland) spectroscopy were located ∼ 1 m above the top of the chamber roof in the exhaust duct (Fig. 2). The FTIR spectrometer measured gaseous emissions prior to dilution to obtain enhanced signal-to-noise ratios for trace gases (Jaakkola et al., 1998). An exhaust gas sample was drawn into the FTIR where the infrared (IR) absorption spectra in the wave number range of 900–4200 cm$^{-1}$ were measured. The instrument software compares the measured absorption spectra with reference gas absorption spectra in the calibration library to identify gas species and calculate concentrations. Examples of reference gas spectra and an Everglades (FL) peat sample spectrum are plotted in Fig. S3.

Smoke from the chamber was drawn through a dilution sampling manifold where the exhaust was diluted with clean air to achieve cooling that allowed for condensation of SVOCs. A portion of the exhaust was directed through a potential aerosol mass (PAM)-OFR (Aerodyne Research Inc., Billerica, MA, USA) to simulate atmospheric aging prior to quantification by the sampling instruments shown in Fig. 2. The 185 and 254 nm (OFR185) ultraviolet (UV) lamps in the OFR were operated at 2 and 3.5 V with 10 L min$^{-1}$ flow rate to simulate intermediate-aged (∼ 2 d) and well-aged (∼ 7 d) emissions, assuming an average daily OH con-

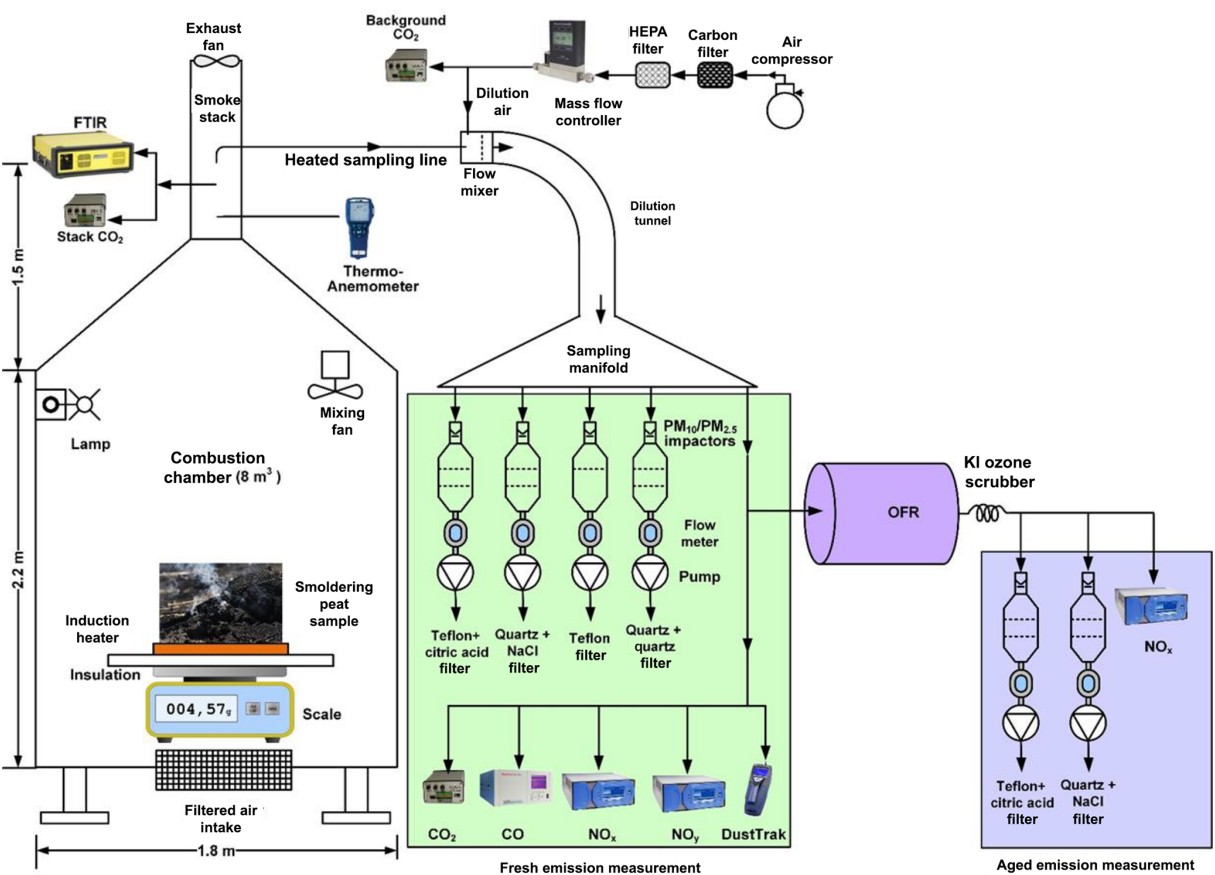

**Figure 2.** Configuration for peat combustion experimental setup. (FTIR: Fourier transform infrared spectrometer; OFR: oxidation flow reactor; OFR lamps were operated at 2 and 3.5 V to simulate aging of $\sim$ 2 and 6.79 d, respectively.)

centration of $1.5 \times 10^6$ molecules cm$^{-3}$. The estimated OH exposures (OH$_{exp}$) at 2 and 3.5 V were $2.6 \times 10^{11}$ and $8.8 \times 10^{11}$ molecules s cm$^{-3}$ based on the measured decay of sulfur dioxide (SO$_2$). Due to external OH reactivity from carbon monoxide (CO), NO$_x$, and other reactants, these OH$_{exp}$ levels represent upper limits of the actual OH exposures inside the OFR (Peng et al., 2015; Li et al., 2015).

Oxides of nitrogen were measured as NO$_x$ (the sum of NO and NO$_2$) and total reactive nitrogen (NO$_y$, including NO, NO$_2$, N$_2$O$_5$, HNO$_3$, HNO$_4$, ClONO$_2$, HONO, alkyl nitrates, and PAN) by chemiluminescence NO$_x$ and NO$_y$ analyzers (Ballentin et al., 2003; Allen et al., 2018). The NO$_x$ analyzers placed upstream and downstream of the OFR determined NO$_x$ changes with OH$_{exp}$ in the OFR. There are known interferences for the nonselective catalytic converter in the chemiluminescent NO$_x$ analyzer and for spectroscopic absorption in the FTIR (Allen et al., 2018; Prenni et al., 2014; Villena et al., 2012). The chemiluminescence monitor converts most nitrogenous compounds to NO, with HNO$_3$ and PAN being the most important potential interferents (Winer et al., 1974). However, much of the available HNO$_3$ and PAN is removed by the tubing leading to the molybdenum converter in the standard NO$_x$ analyzer, which is why the NO$_y$

analyzer locates the converter at the inlet. Allen et al. (2018) found no significant differences between NO$_x$ measurements of biomass burning plumes when comparing a chemiluminescent analyzer with more specific UV absorption measurements.

The following analyses are based on (1) the commercial NO$_x$ analyzers for NO, NO$_2$, and NO$_x$ (NO+NO$_2$ as equivalent NO$_2$); (2) the NO$_y$ analyzer for total reactive nitrogen; and (3) the FTIR spectrometer for trace gas measurements of methane (CH$_4$), NH$_3$, HCN, nitrous oxide (N$_2$O), and 13 low-molecular-weight VOCs (C$_2$–C$_6$).

PM$_{2.5}$ filter packs were sampled upstream and downstream of the OFR to characterize fresh and aged emissions, respectively, with MiniVol PM$_{2.5}$ samplers (Airmetrics, Springfield, OR, USA) operated at 5 L min$^{-1}$ flow rate per channel. PM$_{2.5}$ mass, elements, carbon, water-soluble organic carbon (WSOC), ions, carbohydrates, organic acids, and gaseous NH$_3$ and HNO$_3$ were obtained from the paired upstream and downstream filter samples to examine changes in speciated EFs and source profiles with photochemical aging. Average filter-based EFs are examined by peat types and aging times (i.e., denoted as fresh 2 vs. aged 2 and fresh 7 vs. aged 7) (Chow et al., 2019).

## 2.3 Filter pack measurements

PM$_{2.5}$ mass and major chemical species concentrations were obtained from the parallel Teflon-membrane and quartz-fiber filters (CE2 Teflo©, 2 µm pore size, R2PJ047 and Tissuquartz 2500 QAFUP, Pall Life Sciences, Port Washington, NY, USA). Teflon-membrane filters were equilibrated in a temperature (20–23 °C) and relative humidity (30 %–40 %) controlled environment for a minimum of 48 h prior to gravimetric analysis by a microbalance with ±1 µg sensitivity (Watson et al., 2017). This was followed by multielemental analysis by X-ray fluorescence (Watson et al., 1999). Quartz-fiber filters were prefired at 900 °C for 4 h to minimize organic artifacts. A portion (0.5 cm$^2$) of the quartz-fiber filter was submitted for organic, elemental, and brown carbon (OC, EC, and BrC) analysis following the IMPROVE_A thermal/optical reflectance (TOR) protocol (Chow et al., 2007, 2015). Half of the quartz-fiber filters was extracted in DDW for ionic speciation (i.e., chloride, Cl$^-$; nitrate, NO$_3^-$; nitrite, NO$_2^-$; sulfate, SO$_4^=$; water-soluble sodium, Na$^+$, and potassium, K$^+$; ammonium, NH$_4^+$; 17 carbohydrates; and 10 organic acids) by ion chromatography (Chow and Watson, 2017) and for WSOC by combustion and nondispersive infrared detection. Citric acid and sodium chloride impregnated cellulose-fiber filters placed behind the Teflon-membrane and quartz-fiber filters, respectively, acquired NH$_3$ as NH$_4^+$ and HNO$_3$ as volatilized nitrate, respectively, with analysis by ion chromatography. Details on chemical analyses can be found in Chow et al. (2019).

The open face sampling manifold allows homogenous particle deposits on 47 mm filters (Watson et al., 2017). To test the uniformity of particle deposits, five individual punches were removed from the center and each quadrant of the 47 mm quartz-fiber filter disc for carbon analyses. Table S2 shows total carbon (TC = OC + EC) concentration variations of 1.7 % to 5 % across the filters for the five test burns, within the overall uncertainty of the emission estimates. Standard deviations from the five filter punches for each experiment are low with coefficients of variation of 1.7 %–5.0 %.

## 2.4 Modified combustion efficiency and fuel-based emission factors

The modified combustion efficiency (MCE) is defined as the ratio of background-subtracted CO$_2$ to the sum of CO$_2$ and CO (Ward and Radke, 1993):

$$\text{MCE} = \frac{\Delta\text{CO}_2}{\Delta\text{CO}_2 + \Delta\text{CO}}, \tag{1}$$

where $\Delta\text{CO}_2$ and $\Delta\text{CO}$ are CO$_2$ and CO concentrations above background. MCE provides a real-time indicator of the combustion status (e.g., MCE $> \sim 0.9$ for flaming and MCE $< \sim 0.85$ for smoldering).

Each burn was completed when concentrations of pollutants measured online (i.e., CO, NO$_x$, NO$_y$, and PM$_{2.5}$) returned to the baseline/background levels. Dilution ratios ranging from 2.7 to 5 were taken into account when calculating EFs. Fuel-based EFs are calculated based on carbon mass balance, expressed as grams of emission per kilogram of dry fuel (g kg$^{-1}$) (Wang et al., 2012). For gaseous and particle species $i$, the time-integrated EF$_i$ is

$$\text{EF}_i = \text{CMF}_{\text{fuel}} \frac{C_i}{\left[ \begin{array}{c} C_{\text{CO}_2}\left(\frac{M_\text{C}}{M_{\text{CO}_2}}\right) + C_{\text{CO}}\left(\frac{M_\text{C}}{M_{\text{CO}}}\right) + C_{\text{CH}_4}\left(\frac{M_\text{C}}{M_{\text{CH}_4}}\right) \\ + \sum_j C_{\text{VOC}_j}\left(\frac{n_j \times M_\text{C}}{M_{\text{VOC}_j}}\right) + \text{PM}_\text{c} \end{array} \right]} \times 1000, \tag{2}$$

where CMF$_{\text{fuel}}$ is the carbon mass fraction of the fuel in kilograms of carbon per kilogram of fuel; $C_i$, $C_{\text{CO}_2}$, $C_{\text{CO}}$, $C_{\text{CH}_4}$, and C$_{\text{VOC}_j}$ are the background-subtracted concentrations for species $i$ (e.g., nitrogen or PM$_{2.5}$ species), CO$_2$, CO, CH$_4$, and VOC (C$_2$–C$_6$) species $j$ in milligrams per cubic meter (mg m$^{-3}$) under standard conditions (temperature $= 293$ K and pressure $= 1$ atm), respectively; PM$_c$ is the total carbon concentration of PM$_{2.5}$ in milligrams per cubic meter (mg m$^{-3}$); $M_\text{C}$, $M_{\text{CO}_2}$, $M_{\text{CO}}$, $M_{\text{CH}_4}$, and $M_{\text{VOC}_j}$ are the atomic or molecular weights of carbon, CO$_2$, CO, CH$_4$, and VOC species $j$ in milligrams per mole, respectively; $n_j$ is the number of carbon atom in VOC species $j$; and the factor 1000 converts kilograms to grams. All concentrations are converted to stack concentration; i.e., species measured after dilution are adjusted by the dilution ratio. Equation (2) assumes that the carbon mass in unmeasured VOCs and other emissions not listed above is negligible compared to that in CO$_2$, CO, CH$_4$, measured VOCs (C$_2$–C$_6$), and PM$_{2.5}$ carbon.

## 2.5 Estimation of wall losses

Gas and particle wall losses can result in some underestimation of measured EFs, but it is well within the measurement uncertainties of ±15 %. Losses can occur inside the combustion chamber, in the exhaust stack, sampling lines, and inside the OFR. Due to the low surface-to-volume ratio of the chamber (2.9 m$^{-1}$) and short residence time ($\sim 3$ min) in this study, the gas and particle losses are expected to be low in the combustion chamber. Grosjean (1985) estimated an NH$_3$ loss rate of 4–17 $\times 10^{-4}$ min$^{-1}$ in a small Teflon chamber (3.9 m$^3$) with a surface-to-volume ratio of 3.8 m$^{-1}$, resulting in $< 0.5$ % NH$_3$ wall loss. Even though the NH$_3$ accommodation coefficient might be higher for aluminum than Teflon (Neuman et al., 1999), the chamber wall loss in this study is expected to be $< 5$ % for NH$_3$. To reduce wall losses of sticky gases, the FTIR sampled exhaust gas from the stack without dilution, as shown in Fig. 2. Approximately 9 % NH$_3$ would encounter the stack wall due to turbulent diffusion (Hinds, 1999). The maximum NH$_3$ loss in the stack is $< 9$ %, and the maximum overall NH$_3$ loss is $< 14$ %. Losses of less sticky gases would be lower.

The particle wall loss rates by McMurry and Grosjean (1985) and Wang et al. (2018) indicate $< 5$ % particle

number losses for 10 nm–2.5 µm in a similar chamber. Particle losses by turbulent diffusion in the stack are also low (< 0.5 %). For a 2 m long horizontal heated sampling line in this study (Fig. 2), particle losses by diffusion and gravitational settling are negligible (< 0.1 %) for 10 nm–1 µm particles and ~ 6 % for 2.5 µm particles. Earlier measurements showed that the dilution tunnel had ~ 100 % penetration for 0.5–5 µm particles (Wang et al., 2012). Therefore, maximum particle losses in this study are estimated to be < 5 % for 10 nm–1 µm and < 10 % for 2.5 µm. Past studies (Lambe et al., 2011; Bhattarai et al., 2018; Karjalainen et al., 2016) showed that particle number losses through the OFR may be ~ 50 % for 20 nm and < 10 % for > 100 nm particles, with a negligible effect on mass concentration.

## 3   Results and discussion

### 3.1   Fuel composition

Table 1 shows that peat contains 44 % C–57 % C and 31 % O–39 % O with the exception of the two Guizhou, China, peats (20 % C–30 % C and 21 % O–24 % O). The carbon content (50.6 ± 2.5 % C) in the Borneo, Malaysia, peat is within the range of carbon fractions reported for the Kalimantan and Sumatra, Indonesia, peat (44 % C–60 % C) (Christian et al., 2003; Hatch et al., 2015; Iinuma et al., 2007; May et al., 2014; Setyawati et al., 2017). The low carbon content (20 % C–30 % C) of Guizhou peats is similar to the 28 % C–30 % C reported for two eastern North Carolina, USA, peats (Black et al., 2016).

Hydrogen contents of 2 % H–7 % H in Table 1 are consistent with abundances found elsewhere, including (1) ~ 6 % H for northern Minnesota, USA, peat (Yokelson et al., 1997); (2) ~ 2 % H–3 % H for the eastern North Carolina peat (Black et al., 2016); and (3) ~ 5 % H–7 % H for Indonesian peats (Iinuma et al., 2007; Christian et al., 2003; Hatch et al., 2015). Sulfur (S) contents are below detection limits (< 0.01 %), and nitrogen contents are 1 % N–4 % N. Ratios of N/C are 0.02–0.08, consistent with the reported N/C ratios of (1) 0.036 for Neustädter Moor, northern Germany (Iinuma et al., 2007); (2) 0.017–0.04 for Ireland and the United Kingdom (Wilson et al., 2015); (3) 0.02–0.03 for Alberta and Ontario, Canada (Stockwell et al., 2014); (4) 0.062 for Minnesota, USA (Yokelson et al., 1997); (5) 0.022–0.03 for the eastern coast of North Carolina, USA (Black et al., 2016); and (6) 0.036–0.039 for Kalimantan and Sumatra, Indonesia (Christian et al., 2003; Hatch et al., 2015).

The sum of elements (i.e., C, H, N, S, and O) accounts for 91 %–98 % of total mass except for the Guizhou peats (47 %–56 %). As Guizhou peats appear to be a mixture of peat and soil, these samples may represent degraded peats (Miettinen et al., 2017) or contain additional minerals or high ash contents, similar to North Carolina peats (44 %–62 % ash, Black et al., 2016). Therefore, these peats were only used for pre-

liminary testing of sample ignition and heating to optimize burning conditions. Overall, the six other peats in Table 1 represent biomes from different regions of the world.

### 3.2   Emission factors (EFs)

Table S3 summarizes the 40 peat combustion tests with the peat masses before and after each burn. The afterburn residue may have contained unburned peat as well as noncombustible ash. The residues were not analyzed for carbon and nitrogen contents. A few samples were voided due to sampling abnormalities. The following analyses are based on the 32 paired (fresh vs. aged) samples at 25 % fuel moisture and 3 paired samples at 60 % fuel moisture. The amount of fuel consumed per test ranged from 21 to 48 g for all but Russian peat (14–15 g) due to limited supply.

PM$_{2.5}$ mass concentrations, in the range of 328–2277 µg m$^{-3}$, are 1 to 2 orders of magnitude higher than those commonly measured at ambient monitoring sites. Typical sample durations from ignition to completion were ~ 40–60 min, except for the Everglades (FL) peats that took longer (up to 135 min). Similar particle loadings (mostly within ± 20 %) were found for downstream (aged) and upstream (fresh) samples. The exception is Everglades (FL) peat, where prolonged sample durations and 7 d aging times resulted in higher downstream particle loadings with ratios of aged/fresh mass concentrations ranging from 1.6 to 2.0.

#### 3.2.1   Gaseous carbon emission factors

Individual and average carbonaceous gas EFs are summarized in Table S4. As shown in Fig. S4, variations by biome are found among the different peats with relative standard deviations ranging from 2 % to 27 %. The largest EFs are found for CO$_2$ (EF$_{CO_2}$), ranging from 994 to 1455 g kg$^{-1}$, which are 1–2 orders of magnitude higher than the corresponding EF$_{CO}$ and EF$_{CH_4}$. Average EF$_{CO_2}$ varied by > 30 % among biomes, ranging from 1073 ± 61 to 1400 ± 38 g kg$^{-1}$ for the Russian and Alaskan peats, respectively.

Muraleedharan et al. (2000) reported the first laboratory-combustion EFs of 150–185 for EF$_{CO_2}$, 15–37 for EF$_{CO}$, and 6–11 g kg$^{-1}$ for EF$_{CH_4}$ on a wet mass basis for Brunei peat with a 51.4 % moisture content. Table 2 shows studies conducted over the past decade, with more field monitoring during the 2015 El Niño–Southern Oscillation (ENSO) period in Indonesia. Open path (OP)-FTIR was commonly used to acquire gaseous emissions with MCEs ranging from 0.77 to 0.86, consistent with smoldering combustion. A limited number of burns ($n$ of 1 to 6) were conducted in laboratories using combustion chambers, whereas a larger number of in situ field-burn samples ($n$ of 17 to 35) were acquired for southeast Asian peats (Wooster et al., 2018; Setyawati et al., 2017; Stockwell et al., 2016).

Table 2 exhibits > 2-fold variations in EF$_{CO_2}$ among studies. The highest EF$_{CO_2}$ with the lowest variability was found

Please note the remarks at the end of the manuscript.

**Table 1.** Average peat composition* (dry weight percentage) for total carbon (C), hydrogen (H), nitrogen (N), sulfur (S), and oxygen (O).

| Peat location | C (%) | H (%) | N (%) | S (%) | O (%) | N/C mass ratio | Sum (CHNSO; %) |
|---|---|---|---|---|---|---|---|
| Odintsovo, Russia | 44.20 ± 1.01 | 6.43 ± 0.16 | 1.50 ± 0.52 | < 0.01 | 38.64 ± 0.78 | 0.034 | 90.8 |
| Pskov, Siberia | 52.03 ± 0.23 | 6.30 ± 0.05 | 2.92 ± 0.12 | < 0.01 | 36.83 ± 0.39 | 0.056 | 98.1 |
| Northern Alaska, USA | 50.94 ± 0.81 | 6.05 ± 0.07 | 1.79 ± 0.09 | < 0.01 | 36.62 ± 0.30 | 0.035 | 95.4 |
| Putnam County Lakebed, Florida, USA | 56.64 ± 0.37 | 6.25 ± 0.40 | 3.53 ± 0.05 | < 0.01 | 31.43 ± 0.36 | 0.062 | 97.8 |
| Everglades, Florida, USA | 47.22 ± 0.57 | 5.15 ± 0.16 | 3.93 ± 0.08 | < 0.01 | 34.18 ± 0.87 | 0.083 | 90.5 |
| Caohai, Guizhou, Southeast China | 19.74 ± 2.01 | 2.09 ± 1.26 | 1.35 ± 0.16 | < 0.01 | 23.95 ± 1.15 | 0.068 | 47.1 |
| Gaopo, Guizhou, Southeast China | 29.70 ± 2.09 | 3.13 ± 0.16 | 2.08 ± 0.22 | < 0.01 | 21.46 ± 1.27 | 0.070 | 56.4 |
| Borneo, Malaysia | 50.55 ± 2.53 | 6.46 ± 0.99 | 1.16 ± 0.08 | < 0.01 | 33.72 ± 0.30 | 0.023 | 91.9 |

* Elemental analyses were performed using an elemental analyzer (Flash EA1112 CHNS/O Analyzer, Thermo Fisher Scientific, Waltham, MA, USA). Each dried peat sample (∼ 2–3 g) was submitted for combustion analysis at 900 °C for C, H, N, and S in a helium/oxygen atmosphere and at 1060 °C for O in a helium atmosphere. Three to four replicate sample analyses were conducted for each type of peat to obtain the average and standard deviations.

for tropical peats (ranges 1331–1831 g kg$^{-1}$ for smoldering). Average EF$_{CO_2}$ (1331 ± 78 g kg$^{-1}$) for Malaysian peat ($n = 6$) from this study is ∼ 16 % and ∼ 18 % lower than the 1579 ± 58 and 1615 ± 184 g kg$^{-1}$ for Peninsula, Malaysia (Smith et al., 2018), and average boreal/temperate peats (Hu et al., 2018), respectively. Malaysian peat EF$_{CO_2}$ measured in this study is 20 % lower than the 1681 ± 96 g kg$^{-1}$, averaged from seven studies of Kalimantan and Sumatra, Indonesia, peats (Christian et al., 2003; Stockwell et al., 2014; Huijnen et al., 2016; Nara et al., 2017).

Overall average EF$_{CO_2}$ values (1269 ± 139 g kg$^{-1}$, $n = 32$) from this study (Table S4) are ∼ 19 %–25 % lower than the 1563 ± 65 g kg$^{-1}$ for peatland fires used in atmospheric models (Akagi et al., 2011), 1550 ± 130 g kg$^{-1}$ in a recent review (Andreae, 2019), and 1703 g kg$^{-1}$ (Christian et al., 2003) adopted by the 2014 Intergovernmental Panel on Climate Change (IPCC) for organic soil fire inventories (IPCC, 2014). EFs derived from this study cover four biomes which may improve global emission estimates.

Average EF$_{CO}$ is typically ∼ 12 %–15 % of EF$_{CO_2}$ in the range of 157–171 g kg$^{-1}$ for all but the two Florida peats with 394 ± 46 g kg$^{-1}$ (MCE = 0.65 ± 0.04) and 93 ± 21 g kg$^{-1}$ (MCE = 0.90 ± 0.03) for the Putnam and Everglades peats, respectively (Tables S4 and 2). This is consistent with a higher EF$_{CO}$ under lower MCEs reported by Setyawati et al. (2017) – a 45-fold increase from 3.1 ± 7.2 g kg$^{-1}$ for flaming (MCE = 0.998 ± 0.005) to 138 ± 72 g kg$^{-1}$ for smoldering (MCE = 0.894 ± 0.055) combustion.

Average EF$_{CO}$ values of 157–161 g kg$^{-1}$ for boreal and temperate peats are ∼ 10 % lower than the 179 ± 61 g kg$^{-1}$ from Hu et al. (2018). The overall average EF$_{CO}$ of 175 ± 92 g kg$^{-1}$ from this study is ∼ 4 % lower than the 182 ± 60 g kg$^{-1}$ in Akagi et al. (2011), ∼ 30 % lower than the 250 ± 23 g kg$^{-1}$ in Andreae (2019), and ∼ 15 % lower than the 207–210 g kg$^{-1}$ used in IPCC (2014).

Average EF$_{CH_4}$ is ∼ 0.3 %–0.9 % of EF$_{CO_2}$, lowest for cold climates with 3.2–6.9 g kg$^{-1}$ for the boreal and temperate peats and 6.7–10.4 g kg$^{-1}$ for the subtropical and tropical peats (Table S4). Table 2 shows that EF$_{CH_4}$ values for Malaysian and Indonesian peats exceed ∼ 10 g kg$^{-1}$ in five of the eight past studies. These EFs are more in line with the 11.8 ± 7.8 in Akagi et al. (2011), 9.3 ± 1.5 in Andreae (2019), and 9–21 g kg$^{-1}$ in IPCC (2014) but are higher than the average (6.6 ± 2.4 g kg$^{-1}$) found in this study.

Emission factors depends on both fuel composition and combustion conditions. Figure S5a shows that total measured gas and particle carbon increases with fuel carbon content for the six types of peat. EF$_{CO_2}$ increases with fuel carbon content (Fig. S5b) except for the Putnam (FL) peat, which has the highest fuel carbon (56.6 %) but low EF$_{CO_2}$. It has high EF$_{CO}$ and EF$_{TC}$ (Fig. S5c–d), consistent with its low MCE (0.65 ± 0.04). EF$_{CO}$ and EF$_{TC}$ do not show a clear trend with fuel carbon content; however, EF$_{CH_4}$ increases with fuel carbon (Fig. S5e) but decreases with fuel oxygen content (Fig. S5f).

### 3.2.2 Gaseous nitrogen emission factors

Individual and average gaseous nitrogen species EFs are summarized in Table S5. EF$_{NO}$ and EF$_{NO_2}$ (Fig. S6b) are low in the range of 0.2–2.1 g kg$^{-1}$. For fresh emissions, most of the NO$_x$ (NO + NO$_2$) is present as NO. After the OFR, NO decreased while NO$_2$ increased, as shown in Fig. S7. A low correlation coefficient ($r = 0.67$) between the downstream and upstream EF$_{NO_x}$ suggests the changes of NO/NO$_2$ ratios between the fresh and aged emissions as well as variabilities among tests.

Table 3 shows that most studies do not report EF$_{NO}$ or EF$_{NO_2}$, partially due to the low concentrations and large variabilities under atmospheric aging. Stockwell et al. (2016, 2014) reported 0.31–1.85 g kg$^{-1}$ EF$_{NO}$ and 2.31–2.36 g kg$^{-1}$ EF$_{NO_2}$ for Indonesia peats. These levels are much higher than

| Sampling location or review (reference) | Sampling method (no. of samples)[b] | Modified combustion efficiency (MCE) | Measurement method | Average emission factors (g kg$^{-1}$) | | | Ratio (EF$_{CO}$ / EF$_{CO_2}$) |
|---|---|---|---|---|---|---|---|
| | | | | EF$_{CO_2}$ | EF$_{CO}$ | EF$_{CH_4}$ | |
| **Boreal** | | | | | | | |
| Odintsovo, Russia (this study) | Lab (n = 6, 25 % FM[c]) | 0.81 ± 0.03 | CO/CO$_2$ monitors and FTIR[d] | 1073 ± 63 | 157 ± 24 | 3.20 ± 0.69 | 0.15 |
| Pskov, Siberia (this study) | Lab (n = 7, 25 % FM[c]) | 0.85 ± 0.01 | CO/CO$_2$ monitors and FTIR[d] | 1380 ± 27 | 159 ± 14 | 6.94 ± 1.48 | 0.12 |
| Western Siberia Chakrabarty et al. (2016) | Lab (n = 1, 25 % FM[c]) (n = 1, 50 % FM[c]) | Smoldering | CO/CO$_2$ monitors | 1432 1698 | 204 49 | NA | 0.14 0.029 |
| **Temperate** | | | | | | | |
| Northern Alaska, USA (this study) | Lab (n = 6, 25 % FM[c]) | 0.86 ± 0.03 | CO/CO$_2$ monitors and FTIR[d] | 1400 ± 38 | 161 ± 19 | 5.69 ± 1.07 | 0.12 |
| Northern Alaska, USA Chakrabarty et al. (2016) | Lab (n = 1, 25 % FM[c]) (n = 1, 50 % FM[c]) | Smoldering | CO/CO$_2$ monitors | 1238 1598 | 83 128 | NA | 0.067 0.08 |
| Hudson Bay lowland, Ontario, Canada Stockwell et al. (2014) | Lab | 0.81 ± 0.009 | FTIR | 1274 ± 19 | 197 ± 9 | 6.25 ± 2.17 | 0.15 |
| Alaska and Minnesota, USA Yokelson et al. (1997) | Lab | 0.81 ± 0.327 | FTIR | 1395 ± 52[e] | 209 ± 68[e] | 6.85 ± 5.66[e] | 0.15 |
| Edinburgh, Scotland, UK Rein et al. (2009) | Lab | Smoldering | Infrared system | 420 ± 134 | 170 ± 33 | NA | 0.40 |
| Sphagnum moss peat, Ireland Wilson et al. (2015) | Lab (n = 5) | 0.84 ± 0.019 | FTIR | 1346 ± 31 | 218 ± 22 | 8.35 ± 1.3 | 0.16 |
| **Subtropical** | | | | | | | |
| Putnam County Lakebed, FL, USA (this study) | Lab (n = 6, 25 % FM[c]) (n = 3, 60 % FM[c]) | 0.65 ± 0.04 0.72 ± 0.01 | CO/CO$_2$ monitors and FTIR[d] | 1126 ± 89 1262 ± 27 | 394 ± 46 315 ± 10 | 10.42 ± 1.81 9.18 ± 0.26 | 0.35 0.25 |
| Everglades National Park, FL, USA (this study) | Lab (n = 3, 25 % FM[c]) | 0.90 ± 0.03 (mix of flaming and smoldering) | CO/CO$_2$ monitors and FTIR[d] | 1292 ± 80 | 93 ± 21 | 7.65 ± 1.36 | 0.07 |
| Pocosin Lakes NWR[f], NC, USA Geron and Hays (2013) | Field (Feb & Aug 2008) (n = 3) | 0.77–0.83 | CO and Infrared gas monitoring | 1010–1140 | 230–300 | NA | NA |
| Green Swamp Preserve, NC, USA Geron and Hays (2013) | Field (Feb 2009) (n = 8) | 0.80–0.81 | CO and Infrared gas monitoring | 1100–1640 | 10–280 | NA | NA |
| Alligator River (AR) NWR[f], NC, USA Geron and Hays (2013) | Field (May 2011) (n = 8) | 0.79–0.86 | CO and Infrared gas monitoring | 1092–1440 | 125–290 | NA | NA |
| Pocosin Lakes NWR[f], NC, USA Black et al. (2016) | Lab (n = 2) | 0.83 ± 0.02 | CO/CO$_2$ monitors | 922 ± 47 | 122 ± 14 | NA | 0.13 |
| Alligator River NWR[f], NC, USA Black et al. (2016) | Lab (n = 2) | 0.86 ± 0.02 | CO/CO$_2$ monitors | 861 ± 112 | 108 ± 20 | NA | 0.13 |
| **Tropical** | | | | | | | |
| Borneo, Malaysia (this study) | Lab (n = 6, 25 % FM[c]) | 0.85 ± 0.02 | CO/CO$_2$ monitors and FTIR[d] | 1331 ± 78 | 171 ± 22 | 6.65 ± 0.93 | 0.13 |
| Peninsula, Malaysia Smith et al. (2018) | Field (Aug 2015) (n = 10) | 0.80 ± 0.03 | FTIR | 1579 ± 58 | 251 ± 39 | 11 ± 6.1 | 0.16 |
| Central Kalimantan, Indonesia Wooster et al. (2018) | Field (Sep/Oct 2015) (n = 23) | 0.81 ± 0.032 | Cavity-enhanced laser absorption spectrometer and FTIR | 1775 ± 64 | 279 ± 44 | 7.9 ± 2.4 | 0.16 |
| Central Kalimantan, Indonesia[j] Stockwell et al. (2016) | Field (Oct/Nov 2015) (n = 35) | 0.77 ± 0.053 | FTIR | 1564 ± 77 | 291 ± 49 | 9.51 ± 4.74 | 0.19 |
| Central Kalimantan, Indonesia Huijnen et al. (2016) | Field (Oct 2015) | 0.8 | Cavity ring-down spectrometer | 1594 ± 61 | 255 ± 39 | 7.4 ± 2.3 | 0.16 |
| West Kalimantan, Indonesia Setyawati et al. (2017) | Lab (n = 17 each) | Flaming (0.998 ± 0.005) Smoldering (0.89 ± 0.06) | CO/CO$_2$ monitors and gas chromatography | 2088 ± 21 1831 ± 131 | 3.10 ± 7.17 138 ± 72 | 0.14 ± 0.13 17 ± 1.2 | 0.0015 0.075 |
| South Kalimantan, Indonesia Stockwell et al. (2014) | Lab (n = 3) | 0.82 ± 0.065 | FTIR | 1637 ± 204 | 233 ± 72 | 12.8 ± 6.61 | 0.14 |

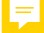

**Table 2.** Peat combustion emission factors (EFs) for $CO_2$, CO, and $CH_4^a$.

| Sampling location or review (reference) | Sampling method (no. of samples)[b] | Modified combustion efficiency (MCE) | Measurement method | Average emission factors (g kg$^{-1}$) | | | Ratio (EF$_{CO}$ / EF$_{CO_2}$) |
|---|---|---|---|---|---|---|---|
| | | | | EF$_{CO_2}$ | EF$_{CO}$ | EF$_{CH_4}$ | |
| South Sumatra, Indonesia Christian et al. (2003) | Lab ($n = 1$) | 0.84 | FTIR | 1703 | 210 | 20.8 | 0.12 |
| North-central Sumatra, Indonesia Nara et al. (2017) | Shipboard (Jun–Aug 2013) ($n = 5$) | 0.84 | Infrared and cavity ring-down spectrometer | $1663 \pm 54$ | $205 \pm 23$ | $7.6 \pm 1.6$ | 0.12 |
| Reviews[g] | | | | | | | |
| Atmospheric modeling Akagi et al. (2011) | NA | NA | NA | $1563 \pm 65$ | $182 \pm 60$ | $11.8 \pm 7.8$ | 0.12 |
| Boreal/temperate | | | | $1327 \pm 150^h$ | $207 \pm 70^h$ | $9 \pm 4^h$ | NA |
| Tropical IPCC (2014) | NA | NA | NA | $1703^i$ | $210^i$ | $21^i$ | NA |
| Boreal/temperate | NA | Smoldering | NA | $1134 \pm 139$ | $179 \pm 61$ | $8.1 \pm 4.1$ | 0.16 |
| Tropical Hu et al. (2018) | | | | $1615 \pm 184$ | $248 \pm 50$ | $12.3 \pm 5.0$ | 0.40 |
| Peat fire Andreae (2019) | NA | NA | NA | $1550 \pm 130$ | $250 \pm 23$ | $9.3 \pm 1.5$ | 0.45 |

[a] Data acquired from this study are so designated. [b] Only included number of samples reported. [c] FM; fuel moisture content. [d] FTIR: Fourier transform infrared spectroscopy. $CH_4$ was acquired by FTIR in this study. [e] Obtained from Stockwell et al. (2014) as only the ratios of moles compound/total moles carbon detected was reported in Yokelson et al. (1997). [f] NWR: National Wildlife Reserve. [g] Reviews for atmospheric modeling and emission inventory development. [h] From Ward and Hardy (1984); Yokelson et al. (1997, 2013). [i] From Christian et al. (2003) for tropical peats. [j] Detailed volatile organic gas emission factors for one of these samples are reported by Koss et al. (2018).

the EF$_{NO_x}$ (as $_{NO_2}$) of $0.75 \pm 0.10$ g kg$^{-1}$ for Malaysian peat in this study.

Emissions for reactive nitrogen, EF$_{NO_y}$ (as $NO_2$), ranged from 0.61 to 6.3 g kg$^{-1}$ with an average of $2.4 \pm 1.4$ g kg$^{-1}$ (Table S5). EF$_{NO_y} > 2.5$ g kg$^{-1}$ are found for the two Florida peats (Fig. S6c) with an average of $4.3 \pm 1.1$ g kg$^{-1}$ for Everglades, which reports the highest nitrogen content ($3.93 \pm 0.08$ %) among peats (Table 1). Figure S5g shows that EF$_{NO}$ increases with fuel nitrogen content, while EF$_{NO_2}$ is not dependent on fuel nitrogen content (Fig. S5h). Because EF$_{NO}$ is higher than EF$_{NO_2}$, EF$_{NO_x}$ and EF$_{NO_y}$ increase with fuel nitrogen content (not shown). Figure S8 shows that $\sim 74$ % of the NO$_y$ is NO$_x$ with a high correlation coefficient ($r = 0.93$). Nitrogen oxides are typically converted to other oxidized nitrogen species within 24 h after emission (Seinfeld and Pandis, 1998; Prenni et al., 2014). The ratio of NO$_x$/NO$_y$ has been used to infer photochemical aging (Kleinman et al., 2003, 2007; Olszyna et al., 1994; Parrish et al., 1992). The high NO$_x$/NO$_y$ ratios suggest that NO$_x$ had not converted to other reactive nitrogen species in the diluted peat plume.

Nitrous oxide (N$_2$O), an inert form of oxide from nitrogen with an atmospheric lifetime of $\sim 110$ years, commonly emitted from fossil fuel, solid waste fertilizers, and biomass combustion, is a greenhouse gas defined by the U.S. EPA (2016). Table S5 shows that EF$_{N_2O}$ are similar to EF$_{NO_y}$ except for Everglades (FL) peat with low EF$_{N_2O}$ ($1.5 \pm 0.3$ g kg$^{-1}$), in the range of 1.1–4.4 g kg$^{-1}$ and average of $2.0 \pm 0.7$ g kg$^{-1}$. The highest average EF$_{N_2O}$ ($3.6 \pm 0.6$ g kg$^{-1}$) is found for Putnam (FL) peat (Fig. S6c).

Hydrogen cyanide (HCN), a known emission from biomass burning (Li et al., 2000; Stockwell et al., 2014), exhibits $> 7$-fold differences (1.8–14 g kg$^{-1}$) in EF$_{HCN}$ (Table S5). The average EF$_{HCN}$ ($11.5 \pm 2.3$ g kg$^{-1}$) for Putnam (FL) peat is 2- to 5-fold higher than for the other biomes (Fig. S6a). Table 3 shows large EF$_{HCN}$ variations among studies, from $0.73 \pm 0.50$ (Ireland, Wilson et al., 2015) to $5.75 \pm 1.60$ g kg$^{-1}$ (Indonesia, Stockwell et al., 2016). More consistent EF$_{HCN}$ values are found for tropical peats in the range of 3–6 g kg$^{-1}$. Average EF$_{HCN}$ values in this study, $4.7 \pm 3.1$ g kg$^{-1}$, are in line with the $5.0 \pm 4.9$ and $4.4 \pm 1.2$ g kg$^{-1}$ reported by Akagi et al. (2011) and Andreae (2019).

EF$_{NH_3}$ values (0.4–8.3 g kg$^{-1}$) are of the same magnitude as EF$_{HCN}$ (Fig. S6a) and independent of fuel nitrogen content (Fig. S5i) except for the Everglades (FL) peat (9–18 g kg$^{-1}$), which has the highest fuel nitrogen content. Total reduced nitrogen emissions, EF$_{NH_3}$ + EF$_{HCN}$, for the two Florida peats (12–25 g kg$^{-1}$) are $\sim 2$- to 3-fold higher than those for other regions. Table 3 also shows high variabilities in EF$_{NH_3}$ among studies (1–11 g kg$^{-1}$). The overall average of $5.6 \pm 4.8$ g kg$^{-1}$ in this study is consistent with the $4.2 \pm 3.2$ g kg$^{-1}$ in Andreae (2019) but $\sim 50$ % of the $10.8 \pm 12.4$ g kg$^{-1}$ in Akagi et al. (2011). The high standard deviations associated with these averages signify large variabilities among experiments.

Figure S9a shows some difference in EF$_{NH_3}$ determined by FTIR and the impregnated filter, especially at high concentrations. The regression slope shows that EF$_{NH_3}$ by the FTIR was $\sim 22$ % lower than that of filters with a correlation

**Table 3.** Peat combustion emission factors (EFs) for gaseous nitrogen species[a].

| Sampling location (reference) | No. of samples | Average emission factors (g kg$^{-1}$) | | | | | | | | Percent NO$_x$/NO$_y$ |
|---|---|---|---|---|---|---|---|---|---|---|
| | | EF$_{NH_3}^b$ | EF$_{HCN}^b$ | EF$_{NO}^c$ | EF$_{NO_2}^c$ | EF$_{NO_x}$ (as NO$_2$) | EF$_{NO_y}^d$ (as NO$_2$) | EF$_{N_2O}^b$ | EF$_{HONO}$ | |
| **Boreal** | | | | | | | | | | |
| Odintsovo, Russia (this study) | 6 | 0.99 ± 0.47 | 2.45 ± 0.43 | 0.34 ± 0.04 | 0.48 ± 0.11 | 1.01 ± 0.14 | 1.06 ± 0.11 | 1.64 ± 0.32 | NA | 95 ± 6 % |
| Pskov, Siberia (this study) | 7 | 4.65 ± 1.38 | 5.00 ± 0.74 | 0.84 ± 0.12 | 0.42 ± 0.03 | 1.70 ± 0.20 | 2.22 ± 0.27 | 2.29 ± 0.29 | NA | 77 ± 5 % |
| Pskov, Siberia Bhattarai et al. (2018) | 3 | NA | NA | NA | NA | 0.08 ± 0.04[e] | NA | NA | NA | NA |
| **Temperate** | | | | | | | | | | |
| Northern Alaska, USA (this study) | 6 | 2.7 ± 0.62 | 2.33 ± 0.22 | 0.84 ± 0.44 | 0.37 ± 0.13 | 1.67 ± 0.76 | 2.10 ± 0.85 | 1.57 ± 0.16 | NA | 79 ± 9 % |
| Hudson Bay lowland, Ontario, Canada Stockwell et al. (2014) | NA | 2.21 ± 0.24 | 1.77 ± 0.55 | NA | NA | NA | NA | NA | 0.18 | NA |
| Alaska and Minnesota, USA Yokelson et al. (1997) | NA | 8.76 ± 13.76 | 5.09 ± 5.64 | NA | NA | NA | NA | NA | NA | NA |
| Sphagnum moss peat, Ireland Wilson et al. (2015) | 5 | 2.20 ± 0.35 | 0.73 ± 0.50 | NA | NA | NA | NA | NA | NA | NA |
| Coastal Swamp land, NC, USA Stockwell et al. (2014) | NA | 1.87 ± 0.37 | 4.45 ± 3.02 | NA | NA | NA | NA | NA | 8.48 ± 0.05 | NA |
| **Subtropical** | | | | | | | | | | |
| Putnam County Lakebed, FL, USA (this study) | 6 (25 % FM) | 3.2 ± 0.26 | 11.5 ± 2.3 | 1.01 ± 0.33 | 0.35 ± 0.28 | 2.01 ± 0.68 | 2.91 ± 0.34 | 3.57 ± 0.63 | NA | 68 ± 15 % |
| | 3 (60 % FM) | 3.3 ± 0.05 | 11.7 ± 0.3 | 0.71 ± 0.07 | 0.65 ± 0.05 | 1.74 ± 0.15 | 2.39 ± 0.19 | 3.89 ± 0.01 | NA | 73 ± 5 % |
| Everglades National Park, FL, USA (this study) | 6 | 11.9 ± 2.01 | 5.12 ± 1.60 | 1.78 ± 0.31 | 0.83 ± 0.16 | 3.56 ± 0.58 | 4.33 ± 1.10 | 1.46 ± 0.28 | NA | 85 ± 14 % |
| Putnam County Lakebed, FL, USA Bhattarai et al. (2018) | | NA | NA | NA | NA | 0.11 ± 0.05[e] | NA | NA | NA | 73 ± 5 % |
| **Tropical** | | | | | | | | | | |
| Borneo, Malaysia (this study) | 6 | 3.66 ± 0.27 | 2.84 ± 0.44 | 0.26 ± 0.04 | 0.35 ± 0.05 | 0.75 ± 0.10 | 1.07 ± 0.56 | 1.88 ± 0.19 | NA | 81 ± 26 % |
| Peninsula, Malaysia Smith et al. (2018) | | 7.82 ± 4.37 | 3.79 ± 1.97 | NA | NA | NA | NA | NA | NA | NA |
| Central Kalimantan, Indonesia Stockwell et al. (2016) | 35 | 2.86 ± 1.00 | 5.75 ± 1.60 | 0.31 ± 0.36 | NA | NA | NA | NA | 0.208 ± 0.059 | NA |
| South Kalimantan, Indonesia Stockwell et al. (2014) | 3 | 1.39 ± 0.79 | 3.30 ± 0.79 | 1.85 ± 0.56 | 2.36 ± 0.03 | NA | NA | NA | 0.1 | NA |
| Overall extratropical peat Stockwell et al. (2014) | NA | 3.38 ± 3.02 | 3.66 ± 2.43 | 0.51 ± 0.12 | 2.31 ± 1.46 | NA | NA | NA | NA | NA |
| **Reviews[g]** | | | | | | | | | | |
| Atmospheric modeling Akagi et al. (2011) | NA | 10.8 ± 12.4 | 5.0 ± 4.93 | NA | NA | 1.23 ± 0.87[f] | NA | NA | NA | NA |
| Smoldering boreal/ Temperate | | 3.39 ± 6.89 | 3.38 ± 3.21 | NA | 2.31 ± 1.46 | NA | NA | NA | NA | NA |
| Smoldering tropical Hu et al. (2018) | | 8.0 ± 3.04 | 5.24 ± 1.55 | | 2.36 ± 0.03 | | | | | |
| Peat fire Andreae (2019) | 3 | 4.2 ± 3.2 | 4.4 ± 1.2 | | | 1.84[f] (±0.48 to 3.4) | NA | NA | NA | NA |

[a] Data acquired from this study are so designated. [b] Data acquired from Fourier transform infrared (FTIR) spectroscopy for this study. [c] Data acquired from the NO$_x$ instrument upstream of the oxidation flow reactor for this study. [d] Data acquired from the NO$_y$ instrument for this study. [e] Reported as NO$_x$. [f] The reported NO$_x$ as NO was converted to NO$_x$ as NO$_2$ for comparison. [g] Reviews for atmospheric modeling and emission inventory development.

coefficient of 0.76. Variable baselines in the FTIR measurements along with some nitrogen content in the diluted air and breath NH$_3$ (Hibbard and Killard, 2011) in the testing laboratory may have contributed to these variations. The impregnated filter collects all of the NH$_3$ over the sampling period, including amounts that are below the FTIR detection limits, so it is probably better representing the time-integrated EF$_{NH_3}$. Reduction of EF$_{NH_3}$ is most apparent after atmospheric aging in Fig. S9b (slope of 0.11), with 2–14 in fresh emissions and reduced to $\sim 0.5$–3 g kg$^{-1}$ after aging.

### 3.2.3 PM$_{2.5}$ mass and carbon emission factors

Continuous PM$_{2.5}$ from the DustTrak with the factory calibration factor yielded PM$_{2.5}$ EFs 3 to 5 times higher than of those derived from gravimetric analyses, higher than the 2-fold mass differences by Wooster et al. (2018). This discrepancy is well known as the factory calibration uses Arizona road dust with a size distribution that is much coarser than that of biomass burning. Therefore, EF$_{PM_{2.5}}$ is calculated from the filter samples. Chow et al. (2019) present the species abundances in PM$_{2.5}$ mass for this study based on the average fresh and aged profiles, separated by 2 and 7 d photochemical aging times simulated with the OFR (Aerodyne, 2019). The same approach is used in Table S6 to compare fresh and aged particle EFs. Comparisons between combined fresh vs. aged EFs for PM$_{2.5}$ mass, carbon (OC, EC, and TC), and levoglucosan for individual tests are shown in Table S7.

Figure S10 shows that EF$_{PM_{2.5}}$ varies > 4-fold (14–61 g kg$^{-1}$) for different peats without large differences between fresh and aged emissions. EF$_{OC}$ varied from 9 to 44 g kg$^{-1}$ while EF$_{EC}$ (0.00–2.2 g kg$^{-1}$) were low (Table S7). The majority of EF$_{PM_{2.5}}$ values consist of EF$_{OC}$, with average EF$_{OC}$ / EF$_{PM_{2.5}}$ ratios of 52 %–98 % by peat type in fresh emissions, followed by $\sim 14$ %–23 % reductions after aging, with the exception of Putnam (FL) peats (remained at 69 %–70 %).

Reductions of EF$_{OC}$ after $\sim 7$ d of photochemical aging are most apparent ($\sim 7$–9 g kg$^{-1}$) for the boreal peats, with the largest degradation for low-temperature OC1 (evolved at 140 °C during carbon analysis), indicating losses of high-vapor-pressure SVOCs upon aging (Table S6). The two Florida peats exhibit an initial EF$_{OC}$ decrease of $\sim 2$ g kg$^{-1}$ after 2 d aging, but with an increase of 1.8–4.0 g kg$^{-1}$ after 7 d. However, these changes are less than the standard deviations associated with the averages.

EF$_{WSOC}$ varies by 5-fold (3–16 g kg$^{-1}$) with over a $\sim 50$ % increase for the Putnam (FL) and Malaysian peats after 7 d. Average EF$_{WSOC}$ by peat type accounts for $\sim 16$ %–36 % and $\sim 20$ %–62 % of fresh EF$_{PM_{2.5}}$ and EF$_{OC}$, respectively. From 2 to 7 d aging, Fig. S11 shows reduced correlation coefficients ($r$ from 0.86 to 0.76 for PM$_{2.5}$, from 0.88 to 0.84 for OC, and 0.94 to 0.68 for WSOC).

As WSOC is part of the OC, the WSOC / OC ratio can be used to illustrate atmospheric aging. Figure S12 shows that WSOC / OC ratios increased by 6 %–16 % after aging. This is attributed to a combination of oxygenation of the aged organic emissions and the reduction of EF$_{OC}$ (Table S7). The increase in WSOC / OC ratios may also be due to photochemical transformation of primary OC to WSOC and/or formation of water-soluble SOA during atmospheric aging (Aggarwal and Kawamura, 2009; Agarwal et al., 2010).

Table 4 compares filter-based PM mass and carbon from different studies. Since different carbon protocols yield different fractions of OC and EC (Watson et al., 2005), the analytical protocols are listed. Most studies follow either IMPROVE_A TOR (Chow et al., 2007) or NIOSH thermal/optical transmittance (TOT) protocols (NIOSH, 1999). As the transmittance pyrolysis correction (i.e., TOT) accounts for charred OC both on the filter surface and organic vapor within the filter substrate, lower EF$_{EC}$ values are expected as compared to TOR (Chow et al., 2004). To remove the OC and EC split uncertainty, TC to PM mass ratios are listed for comparison. Two studies reported black carbon (BC) from a micro-Aethalometer (Wooster et al., 2018) or a single-particle soot photometer (SP2; May et al., 2014). As BC levels are very low, not many differences can be distinguished between BC and EC.

Most studies report EF$_{PM_{2.5}}$ with a few exceptions for EF$_{PM_{10}}$ (Kuwata et al., 2018; Iinuma et al., 2007) and EF$_{PM_1}$ (May et al., 2014). As most of the PM$_{10}$ is in the PM$_{2.5}$ fraction for biomass combustion, particle size fractions have a minor effect on PM EFs (Geron and Hays, 2013; Hu et al., 2018).

Table 4 shows that the majority of EF$_{PM_{2.5}}$ lies in the range of $\sim 20$–50 g kg$^{-1}$ with the exception of very low EF$_{PM_{2.5}}$ values of 4–8 and 6–7 g kg$^{-1}$ reported by Bhattarai et al. (2018) and Black et al. (2016). These are probably due to low filter mass loadings and limited testing ($n$ of 1 to 3), which may result in large uncertainties in gravimetric mass.

Despite different carbon analysis methods, most EF$_{OC}$ lies in the range of $\sim 5$–30 g kg$^{-1}$ with the exception of EF$_{OC}$ (37 g kg$^{-1}$) for Putnam (FL) and EF$_{OA}$ (organic aerosol, 34.5 g kg$^{-1}$) for Indonesian peat measured by a time-of-flight mass spectrometer (May et al., 2014). Most studies show that EF$_{TC}$ accounts for $\sim 60$ %–85 % of the EF$_{PM_{2.5}}$, with low EF$_{EC}$ (0.02–1.3 g kg$^{-1}$).

EF$_{WSOC}$ values of 6–7 and 4–6 g kg$^{-1}$ for the Alaskan and Malaysian peats from this study are consistent with the 6.7 and 3.1 g kg$^{-1}$ from German and Indonesian peats in Iinuma et al. (2007), respectively. EF$_{Levoglucosan}$ exhibits > 2 orders of magnitude variabilities among the biomes with 0.24–16 and 0.24–9.6 g kg$^{-1}$ in fresh and aged emissions, respectively.

Past studies show that the extent of levoglucosan degradation depends on OH exposure in the OFR, organic aerosol composition, and vapor wall losses (e.g., Bertrand et al., 2018a, b; Hennigan et al., 2010; Hoffmann et al., 2010; May et al., 2012; Lai et al., 2014; Pratap et al., 2019). Potential chemical pathways for the formation of organic species

| Sampling location (reference) | Sampling method (no. of samples) | Modified combustion efficiency (MCE) | Carbon analysis method[b] | Average emission factor (g kg$^{-1}$) | | | |
|---|---|---|---|---|---|---|---|
| | | | | EF[c]$_{PM_{2.5}}$ (PM size) | EF$_{OC}$ | EF$_{EC}$ | Ratio (EF$_{TC}$ / EF$_{PM}$) |
| **Boreal** | | | | | | | |
| Odintsovo, Russia (this study)[a] | Lab (n = 6, 25 % FM)[d] | 0.81 ± 0.03 | IMPROVE_A | 42.6 ± 5.2 (fresh)[e] 40.5 ± 7.2 (aged)[e] | 25.1 ± 3.3 (fresh)[e] 17.2 ± 2.7 (aged)[e] | 0.77 ± 0.38 (fresh)[e] 0.69 ± 0.19 (aged)[e] | 0.61 ± 0.05 0.45 ± 0.07 |
| Siberia (this study)[a] | Lab (n = 7, 25 % FM)[d] | 0.85 ± 0.01 | IMPROVE_A | 33.9 ± 6.3 (fresh)[e] 30.7 ± 10.2 (aged)[e] | 26.0 ± 3.4 (fresh)[e] 18.1 ± 4.5 (aged)[e] | 0.69 ± 0.58 (fresh)[e] 0.78 ± 0.31 (aged)[e] | 0.80 ± 0.08 0.64 ± 0.13 |
| Pskov, Siberia Bhattarai et al. (2018) | Lab (n = 3) | NA | IMPROVE_A | 7.98 ± 1.58 | 6.52 ± 1.4 | 0.02 ± 0.01 | 0.82 |
| Western Siberia Chakrabarty et al. (2016) | Lab (n = 1, 25 % FM)[d] (n = 1, 50 % FM)[d] | < 0.7 | IMPROVE_A | NA | 17 11 | 0.2 0.1 | NA |
| Neustädter Moor, Northern Germany Iinuma et al. (2007) | Lab | 0.84 | VDI | 44 (PM$_{10}$)[g] | 12.8 | 0.96 | 0.31 |
| **Temperate** | | | | | | | |
| Northern Alaska, USA (this study)[a] | Lab (n = 6, 25 % FM)[d] | 0.85 ± 0.02 | IMPROVE_A | 24.0 ± 7.6 (fresh)[e] 24.8 ± 5.3 (aged)[e] | 17.4 ± 4.1 (fresh)[e] 14.9 ± 3.9 (aged)[e] | 0.60 ± 0.24 (fresh)[e] 0.55 ± 0.42 (aged)[e] | 0.77 ± 0.12 0.63 ± 0.16 |
| Interior Alaska, USA Chakrabarty et al. (2016) | Lab (n = 1, 25 % FM)[d] (n = 1, 50 % FM)[d] | 0.7 0.7 | IMPROVE_A | NA | 7 4 | 0.1 0.2 | NA |
| **Subtropical** | | | | | | | |
| Putnam County Lakebed, FL, USA (this study)[a] | Lab (n =4, 25 % FM)[d] | 0.65 ± 0.04 | IMPROVE_A | 53.1 ± 6.8 (fresh)[e] 53.9 ± 8.3 (aged)[e] | 36.6 ± 1.9 (fresh)[e] 37.3 ± 6.7 (aged)[e] | 1.33 ± 0.60 (fresh)[e] 0.95 ± 0.07 (aged)[e] | 0.72 ± 0.05 0.71 ± 0.04 |
| | Lab (n = 2, 25 % FM)[d] | 0.67 ± 0.02 | | 51.6 ± 7.9 (fresh 2)[f] 48.2 ± 8.4 (aged 2)[f] | 36.6 ± 1.8 (fresh 2)[f] 34.0 ± 8.3 (aged 2)[f] | 1.8 ± 0.61 (fresh 2)[f] 0.99 ± 0.15 (aged 2)[f] | 0.85 ± 0.04 0.66 ± 0.10 |
| | Lab (n = 3, 60 % FM)[d] | 0.72 ± 0.01 | | 35.9 ± 4.3 (fresh 2)[f] 34.7 ± 2.6 (aged 2)[f] | 29.3 ± 2.2 (fresh 2)[f] 22.1 ± 2.3 (aged 2)[f] | 1.00 ± 0.07 (fresh 2)[f] 0.85 ± 0.85 (aged 2)[f] | 0.75 ± 0.11 0.72 ± 0.08 |
| Everglades National Park, FL, USA (this study)[a] | Lab (n = 7, 25 % FM)[d] | 0.90 ± 0.03 | IMPROVE_A | 23.6 ± 5.1 (fresh)[e] 33.5 ± 11.4 (aged)[e] | 19.0 ± 4.4 (fresh)[e] 18.8 ± 5.2 (aged)[e] | 0.78 ± 0.45 (fresh)[e] 0.67 ± 0.30 (aged)[e] | 0.85 ± 0.15 0.60 ± 0.12 |
| Pocosin Lakes NWR[h], NC, USA Geron and Hays (2013) | Field (n = 3) (Feb & Aug 2008) | 0.77–0.83 | NA | 34–55 | NA | NA | NA |
| Green Swamp Preserve, NC, USA Geron and Hays (2013) | Field (n = 8) (Feb 2009) | 0.80–0.81 | NA | 44–53 | NA | NA | NA |
| Alligator River NWR[h], NC, USA Geron and Hays (2013) | Field (n = 8) (May 2011) | 0.79–0.86[i] | NA | 48–79 | NA | NA | NA |
| Pocosin Lakes NWR[h], NC, USA Black et al. (2016) | Lab (n = 2) | 0.83 ± 1.02 | NIOSH | 5.9 ± 6.7 | 4.3 ± 4.1 | 0.082 ± 0.091 | 0.74 |
| Alligator River NWR[h], NC, USA Black et al. (2016) | Lab (n = 2) | 0.86 ± 0.02 | NIOSH | 7.1 ± 5.6 | 6.3 ± 4.1 | 0.052 ± 0.057 | 0.89 |
| Putnam County Lakebed, FL, USA Bhattarai et al. (2018) | Lab (n = 3) | NA | IMPROVE_A | 6.89 ± 1.28 | 6.56 ± 1.10 | 0.04 ± 0.02 | 0.96 |
| **Tropical** | | | | | | | |
| Borneo, Malaysia (this study)[a] | Lab (n =4, 25 % FM)[d] | 0.83 ± 0.03 | IMPROVE_A | 22.6 ± 3.1 (fresh)[e] 22.6 ± 5.0 (aged)[e] | 18.0 ± 2.0 (fresh)[e] 14.4 ± 1.7 (aged)[e] | 0.28 ± 0.11 (fresh)[e] 0.29 ± 0.20 (aged)[e] | 0.81 ± 0.02 0.68 ± 0.16 |
| Borneo, Malaysia Bhattarai et al. (2018) | Lab (n = 1) | NA | IMPROVE_A | 3.9 | 9.62 | 0.1 | 2.4 |
| Selangor, Malaysia Roulston et al. (2018) | Field (n = 6) (Jul/Aug 2016) | 0.8–0.85 | NA | 28.0 ± 18.0 | NA | NA | NA |
| Sumatra, Indonesia Christian et al. (2003) | Lab (n = 1) | Smoldering | Unspecified | NA | 6.02 | 0.04 | NA |
| Southern Sumatra, Indonesia Iinuma et al. (2007) | Lab | Smoldering | VDI | 33.0 (PM$_{10}$)[g] | 8 | 0.57 | 0.26 |
| Riau, Indonesia Kuwata et al. (2018) | Field (Jun 2013) Field (Feb–Mar 2014) | NA | NA | 13.0 ± 2.0 (PM$_{10}$) 19.0 ± 2.0 (PM$_{10}$) | NA | NA | NA |
| Central Kalimantan, (Sep/Oct 2015) Wooster et al. (2018) | Field (n = 23) | 0.81 ± 0.032 | NA | 17.82 ± 6.86 | NA | 0.106 ± 0.043 (BC)[j] | NA |
| Central Kalimantan, Indonesia Jayarathne et al. (2018) | Field (n = 21) (Oct/Nov 2015) | 0.78 ± 0.04 | NIOSH | 17.3 ± 6.0 | 12.4 ± 5.4 | 0.24 ± 0.1 | 0.73 |

**Table 4.** Peat combustion emission factors (EFs) for PM$_{2.5}$ mass and carbon[a].

| Sampling location (reference) | Sampling method (no. of samples) | Modified combustion efficiency (MCE) | Carbon analysis method[b] | Average emission factor (g kg$^{-1}$) | | | |
|---|---|---|---|---|---|---|---|
| | | | | EF$_{PM_{2.5}}$[c] (PM size) | EF$_{OC}$ | EF$_{EC}$ | Ratio (EF$_{TC}$ / EF$_{PM}$) |
| Indonesia (location not specified) May et al. (2014) | Lab | 0.89 | TOF-AMS and SP2 | 34.9 (PM$_1$)[k] | 34.5 (OA)[k] | 0.01 (BC)[k] | 0.99 |
| Reviews[l] | | | | | | | |
| Peatlands from tropical forest Akagi et al. (2011) | NA | NA | NA | NA | 6.23 ± 3.6 | 0.2 ± 0.11 | NA |
| Smoldering Boreal/temperate Smoldering tropical Hu et al. (2018) | NA NA | NA NA | NA NA | 19.2 ± 6.8 17.3 ± 6.0 | 8.38 ± 4.14 8.8 ± 4.24 | 0.36 ± 0.28 0.28 ± 0.18 | 0.46 0.52 |
| Peat fires Andreae (2019) | NA | NA | NA | 17.3 | 12.4 | 0.19 | 0.73 |

[a] Data acquired from this study are so designated. [b] The IMPROVE_A protocol reports OC and EC by thermal/optical reflectance (TOR, Chow et al., 2007); the NIOSH and NIOSH5040 reports OC and EC by thermal/optical transmittance (NIOSH, 1999); VDI is the German Industrial Standard (VDI, 1999); TOF-MS: time-of-flight mass spectrometer (Drewnick et al., 2005); and single-particle soot photometer (SP2, DMT Inc., Boulder, CO, USA) measures black carbon (BC) by laser-induced incandescence technique (Stephens et al., 2003). [c] Size fraction is PM$_{2.5}$ except where otherwise noted. [d] FM; fuel moisture. [e] Includes averages of all fresh and all aged emission factors (EFs) for the 25 % fuel moisture (i.e., grouped fresh 2 and fresh 7 vs. aged 2 and aged 7 shown in Table S7). [f] Comparisons between 25 % and 60 % fuel moisture content are only made with fresh 2 vs. aged 2 of Putnam (FL) peats. [g] Sum of five stages of Berner Impactor with 0.05–0.14, 0.14–0.42, 0.42–1.2, 1.2–3.5, and 3.5–10 µm size ranges. [h] National Wildlife Refuge, eastern NC. [i] From Jayarathne et al. (2018). [j] BC by micro-Aethalometer (AE 51) (Cheng et al., 2013; Wooster et al., 2018). [k] PM$_1$ and organic aerosol (OA) acquired from time-of-flight mass spectrometry (TOF-MS) measurements (Drewnick et al., 2005). [l] Reviews for atmospheric modeling and emission inventory development.

in biomass combustion emissions were proposed by Gao et al. (2003) that suggested the fragmentation of levoglucosan to C$_3$–C$_5$ diacids, followed by oxalic acid, acetic acid, and formic acid. This is consistent with the increases in EF$_{organic\ acids}$ after atmospheric aging, as shown in Table S6. However, detailed chemical mechanisms need to be further investigated.

The highest EF$_{Levoglucosan}$ is found for the fresh Russian peats (15.8 ± 2.9 g kg$^{-1}$), and this is diminished by 45 % after 7 d aging (8.8 ± 2.1 g kg$^{-1}$). Few studies report EF$_{Levoglucosan}$ and results are highly variable. The EF$_{Levoglucosan}$ of 0.57 g kg$^{-1}$ in PM$_{2.5}$ (converted from 46 mg g OC$^{-1}$) by Jayarathne et al. (2018) is ∼ 23 % of the 2.5 g kg$^{-1}$ by Iinuma et al. (2007), both for Indonesia peats. The EF$_{Levoglucosan}$ of 0.5–1.0 g kg$^{-1}$ from fresh Malaysian peat in this study is comparable to 0.57 g kg$^{-1}$ by Jayarathne et al. (2018). The 4.6 g kg$^{-1}$ of EF$_{Levoglucosan}$ for the northern German peat (Iinuma et al., 2007) is higher than the 1.2–4.7 g kg$^{-1}$ found for the average Siberian and Alaskan peats in this study.

EFs for ionic nitrogen species are low (< 0.1 g kg$^{-1}$) in fresh emissions. Both EF$_{NH_4^+}$ and EF$_{NO_3^-}$ increase with 7 d aging – > 0.5 g kg$^{-1}$ EF$_{NH_4^+}$ for all peat and > 1 g kg$^{-1}$ EF$_{NO_3^-}$ for all but Russian (0.79 ± 0.08 g kg$^{-1}$) and Putnam (FL) peats (0.66 ± 0.08 g kg$^{-1}$), consistent with the formation of secondary inorganic aerosol.

### 3.3 Effect of fuel moisture content on emission factors

Only a few studies examine the effects of fuel moisture on peat emissions with inconsistent results. An early study by McMahon et al. (1980) reported high emissions for total suspended particle (TSP, ∼ < 30–60 µm) of 30 ± 20 g kg$^{-1}$ for dry (< 11 % moisture) as compared to 4.1 ± 3.8 g kg$^{-1}$ (after the first 24 h) for wet (53 %–97 % moisture) organic soil. Rein et al. (2009) found higher CO$_2$ (but not CO) yields while increasing fuel moisture to 600 % for tests of boreal Scotland peats in a cone calorimeter which continuously supplies heat to the fuel. Smoldering combustion is possible with high in situ fuel moisture contents when surrounding peat provides insulation and heat from combustion is available for drying just before the advancing front, but such samples will not burn in the laboratory. Watts (2013) sustained lab-based peat smoldering from a cypress swamp (FL) at ∼ 250 % moisture content, which appears to be a maximum.

Table 2 shows that increasing moisture content from ∼ 25 % to ∼ 60 % for the three Putnam (FL) peats resulted in an 11 % increase in EF$_{CO_2}$ but reductions of 20 % EF$_{CO}$ and 12 % EF$_{CH_4}$. No consistent variabilities are found for nitrogen species (Table 3), with negligible changes in EF$_{NH_3}$ and EF$_{HCN}$; a 13 %–30 % reduction in EF$_{NO}$, EF$_{NO_x}$, and EF$_{NO_y}$; and a 45 % increase in EF$_{NO_2}$ and 9 % increase in EF$_{N_2O}$. On the other hand, a reduction of ∼ 30 % EF$_{PM_{2.5}}$ is found (Table 4) as fuel moisture increased from 25 % to 60 %. Higher fuel moisture contents typically result in less efficient burn-

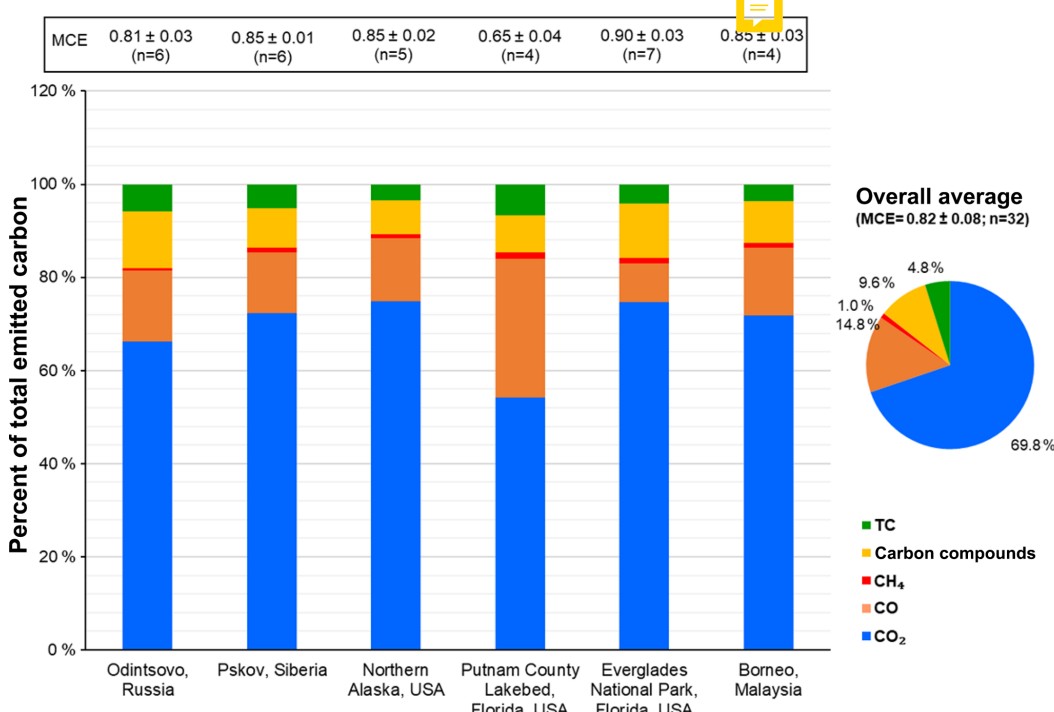

**Figure 3.** [TS3] Average carbonaceous species abundances in total emitted carbon (the sum of carbon in CO$_2$, CO, CH$_4$, ~~VOC~~s, and PM$_{2.5}$ total carbon (TC = OC + EC)). Numbers on top of the bars are average modified combustion efficiencies (MCEs) and the number of samples in each average. The carbon compounds include hydrogen cyanide (HCN), formaldehyde (CH$_2$O), methanol (CH$_3$OH), formic acid (HCOOH), carbonyl sulfide (COS), ethylene (C$_2$H$_4$), ethane (C$_2$H$_6$), acetaldehyde (C$_2$H$_4$O), ethanol (C$_2$H$_5$OH), acetic acid (CH$_3$COOH), propane (C$_3$H$_8$), acrolein (C$_3$H$_4$O), acetone (C$_3$H$_6$O), 3-butadiene (C$_4$H$_6$), benzene (C$_6$H$_6$), hexane (C$_6$H$_{14}$), phenol (C$_6$H$_5$OH), and chlorobenzene (C$_6$H$_5$Cl) acquired by Fourier transform infrared spectrometry.

ing conditions, thereby increasing CO and reducing MCE (Chen et al., 2010). However, an opposite trend is found with EF$_{CO}$ reduced from $394 \pm 46$ to $315 \pm 10$ g kg$^{-1}$ and MCEs increased from $0.65 \pm 0.04$ to $0.72 \pm 0.01$. It is hypothesized that, at higher fuel moisture contents, combustion residence time is slowed enough so that radiant heat transfer from ignited particles to uncombusted areas of peat can be greater, thus increasing the combustion efficiency. It is also possible that the higher water content results in a water–gas shift reaction that converts CO and water to CO$_2$ and hydrogen. Overall, the EFs for $\sim 60$ % moisture contents are comparable to EFs for the six other peats with $\sim 25$ % moisture content.

Increased ($\sim 25$ % to 60 %) fuel moisture yields a $\sim 20$ % reduction in fresh EF$_{OC}$, much lower than the 35 %–43 % reduction ($\sim 25$ % to 50 % moisture) reported by Chakrabarty et al. (2016) for the Siberian and Alaskan peats. By increasing fuel moisture, Chakrabarty et al. (2016) also reported an increase in EF$_{CO_2}$ by 20 % but a $\sim 75$ % reduction and 35 % increase in EF$_{CO}$ for Siberian and Alaskan peats, respectively, based on a single sample.

## 3.4   Distribution of carbon and nitrogen species

Figure 3 shows the distribution of carbonaceous species. Because the EFs are calculated based on the carbon mass balance method (Eq. 2), the total emitted carbon is assumed to be the same as total consumed carbon. The majority ($> 90$ %) of total emitted carbon is present in the gas phase, with 54 % CO$_2$–75 % CO$_2$, followed by 8 % CO–30 % CO. On average, emitted carbon includes $69.8 \pm 7.5$ % CO$_2$, $14.8 \pm 6.5$ % CO, $1.0 \pm 0.3$ % CH$_4$, $9.4 \pm 2.4$ % volatile carbon compounds, and $4.8 \pm 1.3$ % PM$_{2.5}$ TC. The highest ($30 \pm 4$ %) and lowest ($8.4 \pm 1.9$ %) CO abundances for the Putnam (FL) and Everglades (FL) peats are consistent with the lowest and highest average MCEs of 0.65 and 0.90, respectively.

The nitrogen budget in Fig. 4 accounts for 24 %–52 % of nitrogen in the consumed fuel. Since burn temperatures are below those at which NO$_x$ forms from oxygen reactions with N$_2$ in the air; most of the nitrogen in emissions derives from the nitrogen content of the fuels. Kuhlbusch et al. (1991) found N$_2$ emissions constituted an average of $31 \pm 20$ % of nitrogen in consumed grass, hay, pine needle, clover, and wood fuels. Since N$_2$ measurements require

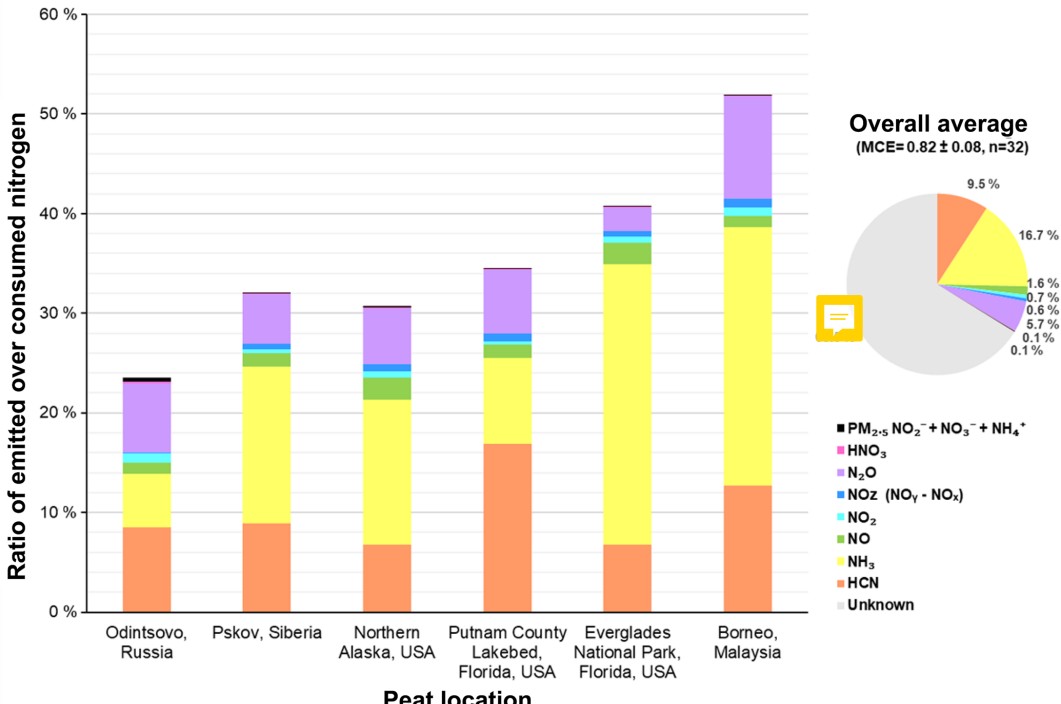

**Figure 4.** TS4 Ratio of emitted over consumed nitrogen for each type of peat (emitted nitrogen is the sum of nitrogen in HCN, NH$_3$, NO, NO$_2$, and NO$_z$ (NO$_y$-NO$_x$), N$_2$O, HNO$_3$, and PM$_{2.5}$ ions (NO$_2^-$ + NO$_3^-$ + NH$_4^+$); and the consumed nitrogen is the product of percent fuel nitrogen content and mass of fuel burned).

combustion in N$_2$-free atmosphere (e.g., a He-O$_2$ mixture), N$_2$ was not quantified here, but it was probably emitted in similar quantities. Isocyanic acid (HNCO) is another important nitrogen-containing compound found in biomass burning emissions (Roberts et al., 2011). Koss et al. (2018) report a 0.16 g kg$^{-1}$ nitrogen-equivalent EF (0.5 g kg$^{-1}$ for HNCO) for a peat sample, comparable to EFs for several of the measured nitrogen compounds summarized in Table 3. Other nitrogen-containing gases reported by Koss et al. (2018) with EFs > 0.1 g kg$^{-1}$ include acetonitrile (CH$_3$CN), acetamide (CH$_3$CONH$_2$), benzonitrile (C$_6$H$_5$CN), and pyridine + pentadienenitriles (C$_5$H$_5$N), which could account for part of the unmeasured nitrogen in emissions. Neff et al. (2002) found that organic nitrogen formed from photochemical reactions of hydrocarbon with NO$_x$ plays an important role in the global nitrogen cycle. Approximately 30 ± 16 % of Neff et al.'s total nitrogen was from organic nitrogen, similar to the 25 % of total nitrogen deposition flux reported by Jickells et al. (2013). Alkaloids, dissolved organic nitrogen, along with nitroaromatic compounds have been reported (e.g., Benitez et al., 2009; Laskin et al., 2009; Kuhlbusch et al., 1991; Koppmann et al., 2005; Kopacek and Posch, 2011; Stockwell et al., 2015).

The majority (> 99 %) of the measured nitrogen in emissions is in the gas phase. On average, 16.7 % of the fuel nitrogen was emitted as NH$_3$ and 9.5 % was emitted as HCN. N$_2$O and NO$_y$ constituted 5.7 % and 2.9 % of nitrogen in the

consumed fuel. NH$_3$ emissions accounted for 26 %–28 % of consumed nitrogen for Everglades (FL) and Malaysian peats, while HCN emissions dominated fuel nitrogen (13 %–17 %) for the Putnam (FL) and Malaysian peats. The fraction of N$_2$O emissions in Malaysian peat nitrogen (10.3 ± 1.1 %) was more than twice the fractions found for the other regions, with reactive nitrogen (NO$_y$) only accounting for 2 %–4 % of the fuel nitrogen. The sum of NH$_3$ and HCN nitrogen ranged from 35 % to 39 % of consumed nitrogen for the Malaysian and Everglades (FL) peats, which is about three times the fraction for Russian peat.

Lobert et al. (1990) point out the importance of nitrogen-containing gases in biomass burning for the atmospheric nitrogen balance. On average, the emitted nitrogen includes 17 ± 10 % NH$_3$, 9.5 ± 3.8 % HCN, 5.7 ± 2.5 % N$_2$O, 2.8 ± 1.0 % NO$_y$ (including NO$_x$), and 0.14 ± 0.18 % of PM nitrogen (sum of NO$_2^-$, NO$_3^-$, and NH$_4^+$). The average nitrogen budget accounts for 35 ± 11 % of the total consumed nitrogen, consistent with past studies showing that around one- to two-thirds of the fuel nitrogen is accounted for during biomass combustion.

## 4   Summary and conclusions

This paper reports fuel composition and emission factors (EFs) from laboratory chamber combustion of six types of

peat fuels representing boreal (Russia and Siberia), temperate (northern Alaska, USA), subtropical (northern and southern Florida, USA), and tropical (Borneo, Malaysia) climate regions. Dried peat fuel contains 44 %–57 % carbon (C), 31 %–39 % oxygen (O), 5 %–6 % hydrogen (H), 1 %–4 % nitrogen (N), and < 0.01 % sulfur (S). The nitrogen to carbon ratios are low, in the range of 0.02–0.08, consistent with peat compositions reported in other studies.

Thirty-two tests with 25 % fuel moisture were reported with predominant smoldering combustion conditions (MCE = 0.82 ± 0.08). Average fuel-based EFs for $CO_2$ ($EF_{CO_2}$) are highest (1400 ± 38 g kg$^{-1}$) and lowest (1073 ± 63 g kg$^{-1}$) for the Alaskan and Russian peats, respectively. $EF_{CO}$ and $EF_{CH_4}$ are ∼ 12 %–15 % and ∼ 0.3 %–0.9 % of $EF_{CO_2}$ in the range of ∼ 157–171 and 3–10 g kg$^{-1}$, respectively. The exception is the two Florida peats, reporting the highest (394 ± 46 g kg$^{-1}$) and lowest (93 ± 21 g kg$^{-1}$) $EF_{CO}$ for Putnam and Everglades, respectively.

Filter-based $EF_{PM_{2.5}}$ varied by > 4-fold (14–61 g kg$^{-1}$) without appreciable changes between fresh and aged emissions. The majority of $EF_{PM_{2.5}}$ consists of $EF_{OC}$, with average $EF_{OC}$ / $EF_{PM_{2.5}}$ ratios by peat type in the range of 52 %–98 % in fresh emissions, followed by ∼ 14 %–23 % reduction after aging with the exception of Putnam (FL) peats (retained at 69 %–70 %). Reduction of $EF_{OC}$ (∼ 7–9 g kg$^{-1}$) are most apparent for boreal peats with the largest decrease in low-temperature OC1 (evolved at 140 °C), suggesting the loss of high-vapor-pressure semivolatile organic compounds during aging. EFs for water-soluble OC ($EF_{WSOC}$) account for ∼ 20 %–62 % of $EF_{OC}$ with ∼ 6 %–16 % increase in $EF_{WSOC}$ / $EF_{OC}$ ratios after aging. The highest $EF_{Levoglucosan}$ is found for Russian peat (15.8 ± 2.9 g kg$^{-1}$) with a 45 % degradation after aging.

The majority (> 90 %) of the total emitted carbon is in the gas phases with 54 %–75 % $CO_2$, followed by 8 %–30 % CO. The nitrogen budget only explains 24 %–52 % of the consumed nitrogen, with an average of 35 ± 11 %, consistent with past studies that around one- to two-thirds of the total nitrogen is lost upon biomass combustion. The majority (> 99 %) of the total emitted nitrogen is in the gas phase, dominated by the two reduced nitrogen species with 16.7 % for $NH_3$ and 9.5 % for HCN. $N_2O$ and $NO_y$ are detectable at 5.7 % and 2.9 % abundance. EFs from this study can be used to refine current emission inventories.

*Data availability.* The data of this study are available from the authors upon request. TS5

*Supplement.* The supplement related to this article is available online at: https://doi.org/10.5194/acp-19-1-2019-supplement.

*Author contributions.* JGW, JCC, JC, L-WAC, and XW jointly designed the study, performed the data analyses, and prepared the manuscript. QW, JT, and SSHH carried out the peat combustion experiments. SG conducted emission factor calculations. ACW acquired peat fuels and provided technical advice on the peat fuel process.

*Competing interests.* The authors declare that they have no conflict of interest.

*Acknowledgements.* This research was primarily supported by the National Science Foundation (NSF, AGS-1464501 and CHE-1214163) as well as internal funding from both the Desert Research Institute, Reno, NV, USA, and Institute of Earth Environment, Chinese Academy of Sciences, Xian, China. The Caohai and Gaopo peat samples were provided by Pinhua Xia of Guizhou Normal University, Guizhou, China, and Chunmao Zhu of the Japan Agency for Marine-Earth Science and Technology, Yokosuka, Japan.

*Financial support.* This research has been supported by the U.S. National Science Foundation (grant nos. AGS-1464501 and CHE-1214163); as well as the National Atmospheric Research Program (2017YFC0212200) and the National Research Program for Key Issues in Air Pollution Control (DQGG0105) in China.

*Review statement.* This paper was edited by James Roberts and reviewed by two anonymous referees.

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

## Remarks from the language copy-editor

## Remarks from the typesetter

**TS1** Please note that all additional remarks have not been inserted. Please note that your comments should be limited to (a) responses to the publisher remarks as given in the manuscript for proofreading, (b) raising Copernicus' awareness of oversights, and (c) expressing disagreement with Copernicus' adjustments. New issues cannot be raised anymore. The probability that an error will occur during proofreading increases with the number of changes. We also wish to avoid having to issue a corrigendum. Thus, we kindly ask you to reduce your corrections to the most relevant ones and in accordance with the guidelines. The proofreading guidelines can be found at: http://publications.copernicus.org/for_ authors/proofreading_guidelines.html. More information on our house standards can be found at: http://publications. copernicus.org/for_authors/manuscript_preparation.html. Especially values cannot be edited at this stage of paper. If you insist on these changes, we will forward your requests to the handling editor for approval. To explain the corrections needed to the editor, please send me the reason why these corrections are necessary. Please note that the status of your paper will be changed to "Post-review adjustments" until the editor has made their decision. We will keep you informed via email.