# Peer review of "Gaseous, PM$_{2.5}$ mass, and speciated emission factors from laboratory chamber peat combustion"

_Atmospheric Chemistry and Physics, 2019_

## Referee Comment (RC1) · Anonymous Referee #1 · 14 Jul 2019

Emission factors from combustion are very important to emission inventory and air quality modeling studies. The authors carried out a series of experiments and reported the gaseous, PM2.5 mass and speciated emission factors from peat combustion. Overall, the experimental methods are reasonable and the data are robust. After the following questions have been well addressed, it is suitable for publishing.

1. Although the authors simply described the apparatus including the combustion chamber and the instruments used in this work. More details should be given, in particular, the characterization details about the combustion chamber. One of my most important concerns is the wall loss for these gaseous and particle phase pollutants in the chamber. Did you consider the wall loss correction when calculating the emission factors?

[Figure]

2. As for C2-C6 VOCs, HCN and NH3 measurements with FTIR, the IR bands for quantifying each species and the details about the quantification methods should be described in the text and tables. It is better to show the typical IR spectrum for these species in the supplement materials. As for NH3, it is a sticky molecule and easily interfered by human activity as pointed out by the authors (line 350). How did you consider these factors on NH3 measurement? In addition, it is necessary to do uncertainty analysis to all of the measured EFs.

3. Most of these results were shown in tables. It is somewhat difficult to follow. For example, when discussing the influence of aging and the moisture of peat on the EFs, it is more easy to understand their differences if showing in figures. In addition, when comparing the measured EFs with the literature results, it is better to discuss the reasons why you obtained a different value. For example, the measured EF($CO_2$) from this work is lower than that in literature. Some objective comments should be given in the text.

4. Even for the same pollutant, EF varied obviously among different samples. Was there a quantitative relationship between the EFs and the element composition in the peat samples or combustion conditions?

5. For OFP experiments, did you consider the OH suppression?

6. In the introduction, the previous relevant researches before this work should be well reviewed.

7. Table S7 was missed.

---

## Author Comment (AC1) · 15 Jul 2019

Good comments that we will be addressing in a detailed revision. Attached is a revision to the supplementary material with Table S7.

Please also note the supplement to this comment: https://www.atmos-chem-phys-discuss.net/acp-2019-456/acp-2019-456-AC1-supplement.pdf

---

## Referee Comment (RC2) · Anonymous Referee #2 · 2 Sep 2019

This manuscript presents measurement results from a laboratory combustion study of peat from different regions around the world. Detailed chemical speciation of both gas-phase and particulate smoke components is provided as a function of fresh vs simulated aged emissions, with a special focus on nitrogen species. The results presented in this paper are helpful to better understand the properties of biomass combustion source emissions and improve emissions inventories. Specifically, the authors presented a very nice detailed, yet concise description of the experimental set-up and procedures. The manuscript is well suited for publication in ACP, and only a few comments and suggestions are given below for the authors to consider in their revised version of the paper.

Specific comments:

[Figure]

1. Line 148-151: What was the rationale for heating the fuels to such a relatively high temperature and reducing the moisture content to such low levels? As the authors point out, some volatile fuel components may have gotten lost during this preparatory step. Has the chosen heating temperature been applied in other similar studies as well?

2. Lines 151-155: The authors may want to add a cautionary statement regarding the re-hydration procedure. According to studies conducted by the USFS Fire Science Lab, it is very difficult to re-hydrate biomass fuels once most of the water has been removed. And even after adding a certain amount of moisture back to the fuel, the physio-chemical properties are not the same as the original ones prior to the drying procedure. Would a potential alternative method be gradual drying to the desired moisture level, and thus maintaining the original water bonding structure?

3. Lines 378-381: Aside from lower OC emission factors, an increase in the WSOC fraction is expected due to the higher degree of oxygenation of the aged organic smoke components, isn't it?

4. Lines 410-411: Could the authors comment on possible degradation pathways that might occur during OFR treatment and potential reaction products of levoglucosan?

5. Lines 433-442: As the authors point out, higher fuel moisture content usually results in lower MCE, and consequently often increases PM emissions. However, the opposite pattern was observed in this study. Could this possibly be related, at least partly, to the re-hydration procedure which may not have restored the original conditions of the wet fuel (see comment No. 2 above)?

6. Lines 487-489: Can the authors add some speculations regarding the "missing" nitrogen, i.e., whether it's due to unidentified nitrogen species, measurement uncertainties, or other reasons?

Technical corrections:

1. Please use consistent spacing between temperature numbers and the degree symbol throughout the manuscript.

2. Line 67-69: Additional field studies by Behera et al. (ESPR, 2014) and Engling et al (ACP, 2014) specifically addresses the effects of peat burning emissions in Southeast Asia.

3. Line 178: Omit the indefinite article at the beginning of the sentence.

4. Line 215: The degree symbol is not needed for temperatures expressed in K units.

5. Line 258: When the authors state the "concentrations were high", it would be helpful for the reader to know a reference point, i.e., "compared to".

6. Lines 313-314: Isn't the low correlation coefficient indicating different emissions characteristics of the fresh vs. aged smoke, and not just a variability between tests?

7. Line 453: Add "volatile" before "carbon".

8. Lines 495-496: The statement regarding the nitrogen content in this sentence is not clear.

9. Lines 510-511: What do the authors mean with "average Russian peat"?

---

## Author Response (AR1)

**Responses to Reviewer #1 Comments**

Emission factors from combustion are very important to emission inventory and air quality modeling studies. The authors carried out a series of experiments and reported the gaseous,  $PM_{2.5}$  mass and speciated emission factors from peat combustion. Overall, the experimental methods are reasonable and the data are robust. After the following questions have been well addressed, it is suitable for publishing.

1. Although the authors simply described the apparatus including the combustion chamber and the instruments used in this work. More details should be given, in particular, the characterization details about the combustion chamber. One of my most important concerns is the wall loss for these gaseous and particle phase pollutants in the chamber. Did you consider the wall loss correction when calculating the emission factors?

**Response (including Parts A and B):**

• **Part A:** The Experimental setup section is reorganized to streamline the flow of description with the addition of the following description (Lines 124-128):

A blower supplied filtered air near the bottom of the chamber. The ventilation rate was controlled by the blower and exhaust fan at ~2.65 m3 min-1, resulting in a smoke residence time of ~ 3 min in the chamber assuming a well-stirred flow model.

Part B: The reported EFs in this study represent the upper limit estimates as wall losses were not corrected. The turbulent diffusion and gravitational settling need to be considered while estimating the wall losses. Theoretically, wall losses can occur inside the combustion chamber, in the exhaust stack, sampling lines, and inside the OFR, but it is difficult to quantify given the variabilities in fuel composition and combustion conditions. For this study, estimated wall losses for the most sticky gas (i.e., NH3) is at most < 14 %, which will be lower for less sticky gases. The particle wall losses depend on particle sizes: < 5 % for 10 nm-1 µm and ~10 % from 2.5 µm particles. Overall, these gaseous and particle wall losses are <15 %, well within the measurement uncertainties of ± 30 % for air quality modeling. A new section (2.5 Estimation of wall losses) is added to address this (Lines 241-264):</li>

Gas and particle wall losses can result in some underestimation of measured EFs, but it is well within the measurement uncertainties of  $\pm 15$  %. Losses can occur inside the combustion chamber, in the exhaust stack, sampling lines, and inside the OFR. Due to the low surface-to-volume ratio of the chamber (2.9 m-1) and short residence time (~3 min) in this study, the gas and particle losses are expected to be low in the combustion chamber. Grosjean (1985) estimated an NH3 loss rate of 4-17 × 10-4 min-1 in a small Teflon chamber (3.9 m3) with a surface-to-volume ratio of 3.8 m-1, resulting in < 0.5 % NH3 wall loss. Even though the NH3 accommodation coefficient might be higher for aluminum than Teflon (Neuman et al., 1999), the chamber wall loss in this study is expected to be < 5 % for NH3. To reduce wall losses of sticky gases, the FTIR sampled exhaust gas from the stack without dilution, as shown in Fig. 2. Approximately 9 % NH3 would encounter the stack wall due to turbulent diffusion (Hinds,

1999). The maximum  $NH_3$  loss in the stack is <9 % and the maximum overall  $NH_3$  loss is <14 %. Losses of less sticky gases would be lower.

The wall loss rates by McMurry and Grosjean (1985) and Wang et al. (2018) indicate <5 % particle number losses for 10 nm–2.5  $\mu$ m in a similar chamber. Particle losses by turbulent diffusion in the stack are also negligible (<0.5 %). For a 2 m-long horizontal heated sampling line in this study (Fig. 2), particle losses by diffusion and gravitational settling are negligible (<0.1 %) for 10 nm - 1  $\mu$ m particles and ~6 % for 2.5  $\mu$ m particles. Earlier measurements showed that the dilution tunnel had ~100% penetration for 0.5-5  $\mu$ m particles (Wang et al., 2012). Therefore, maximum particle losses in this study are estimated to be <5 % for 10 nm - 1  $\mu$ m and <10 % for 2.5  $\mu$ m. Past studies (Bhattarai et al., 2018;Karjalainen et al., 2016;Lambe et al., 2011) showed that particle number losses through the OFR may be ~50 % for 20 nm and <10 % for >100 nm particles, with a negligible effect on mass concentration.

2. As for C2-C6 VOCs, HCN and NH3 measurements with FTIR, the IR bands for quantifying each species and the details about the quantification methods should be described in the text and tables. It is better to show the typical IR spectrum for these species in the supplement materials. As for NH3, it is a sticky molecule and easily interfered by human activity as pointed out by the authors (line 350). How did you consider these factors on NH3 measurement? In addition, it is necessary to do uncertainty analysis to all of the measured EFs.

**Response (including Parts A, B, and C)**:**

• **Part A:** The following description about the FTIR was added (Lines 148-153), along with example spectra of six reference gases (i.e., HCN, NH3, ethene, 1,3-butadiene, hexane, and benzene) and an Everglades (FL) peat sample spectrum in the new Figure S3:

An exhaust gas sample was drawn into the FTIR and the infrared (IR) absorption spectra in the wave number range of  $900 - 4200 \text{ cm}^{-1}$  were measured. The instrument software compares the measured absorption spectra with reference gas absorption spectra in the calibration library to identify gas species and calculate concentrations. Examples of reference gas spectra and an Everglades peat sample spectrum are plotted in Fig. S3.

• **Part B**: To reduce NH3 contamination by background air and human, activities inside the combustion chamber are minimized to only those necessary and the chamber was purged before starting the experiment. During the experiments, the door of the combustion chamber was closed, and intake air passed through a charcoal bed to remove VOCs and a high-efficiency particulate air (HEPA) filter to provide particlefree clean air for dilution. For this study, EFNH3, from an impregnated filter are used as discussed in Section 3.2.2 (Lines 404-410):

Figure S9a shows some scatter in  $EF_{NH_3}$  determined by FTIR and the impregnated filter, especially at high concentrations. The regression slope shows that  $EF_{NH_3}$  by the FTIR was ~22 % lower than that of filters with a correlation coefficient of 0.76. Variable baselines in

the FTIR measurements along with some nitrogen content in the diluted air and breath  $NH_3$  (Hibbard and Killard, 2011) in the testing laboratory may have contributed to these variations. The impregnated filter collects all of the  $NH_3$  over the sampling period, including amounts that are below the FTIR detection limits, so it is probably better representing the time-integrated  $EF_{NH_3}$ .

- **Part C:** For this study, EF uncertainties are reported as standard deviations of multiple runs for each peat. These uncertainties are shown in Tables 2-4, Supplemental Tables S4-S7, Figures S4, S6, and S12.
- 3. Most of these results were shown in tables. It is somewhat difficult to follow. For example, when discussing the influence of aging and the moisture of peat on the EFs, it is more easy to understand their differences if showing in figures. In addition, when comparing the measured EFs with the literature results, it is better to discuss the reasons why you obtained a different value. For example, the measured EF(CO2) from this work is lower than that in literature. Some objective comments should be given in the text.

**Response**:**

- Tables are useful to examine quantitative differences among peat types and to compare with other studies; however, we agree with the Reviewer that graphic presentations are more intuitive for readers. We have added Figure S4 for EFs of CO, CO2, and CH4 based on Table S4; Figure S6 for EFs of NH3, HCN, NO, NO2, NOx, and N2O based on Table S5; Figure S10 for comparison of fresh vs. aged EFs for PM2.5 mass and carbonaceous compounds based on Table 6; and Fig. S12 for comparison of WSOC and OC.
- 4. Even for the same pollutant, EF varied obviously among different samples. Was there a quantitative relationship between the EFs and the element composition in the peat samples or combustion conditions?

**Response**:**

• Indeed, EFs varied by different test runs and by peat types. The linear relationship between the EFs and full elemental composition are more apparent for the sum of gaseous and particulate carbon, and for  $EF_{NO}$ , with inverse association between  $EF_{CH_4}$  and fuel oxygen contents. To illustrate the relationship between EFs and peat fuel content, a new Figure S5 (including Figs. S5a–i) is added to discuss the association between EFs and fuel elemental composition. The following discussions were added to the text:

**• Lines 352-358:**

Emission factors depends on both fuel composition and combustion conditions. Figure S5a shows that total measured gas and particle carbon increases with fuel carbon content for the six types of peat.  $EF_{CO_2}$  increases with fuel carbon content (Fig. S5b) except for the Putnam (FL) peat, which has the highest fuel carbon (56.6%) but low  $EF_{CO_2}$ . It has high  $EF_{CO}$  and  $EF_{TC}$  (Figs. S5c-d), consistent with its low MCE (0.65 ± 0.04).  $EF_{CO}$  and  $EF_{TC}$  do not have a clear

trend with fuel carbon content; however,  $EF_{CH_4}$  increases with fuel carbon (Fig. S5e) but decreases with fuel oxygen content (Fig. S5f).

**• Lines 373-375:**

Figure S5g shows that  $EF_{NO}$  increases with fuel nitrogen content, while  $EF_{NO2}$  is not dependent on fuel nitrogen content (Fig. S5h). Because  $EF_{NO}$  is higher than  $EF_{NO2}$ ,  $EF_{NOx}$  and  $EF_{NOy}$  also increase with fuel nitrogen content (not shown).

**• Lines 396-398:**

 $EF_{NH_3}$  (0.4–8.3 g kg-1) are of the same magnitude as  $EF_{HCN}$  (Fig. S6a) and independent of fuel nitrogen content (Fig. S5i) except for the Everglades (FL) peat (9–18 g kg-1) which has the highest fuel nitrogen content.

5. For OFP experiments, did you consider the OH suppression?

**Response**:**

• The average CO and NOx concentrations upstream of the OFR are 6.3 ppm and 34 ppb, respectively, corresponding to external OH reactivity (OHR) of 45 s-1. The OFR used in this study is a commercial product from Aerodyne and its photon flux ratio at 185nm/254 nm has not been fully characterized (https://sites.google.com/site/pamwiki/hardware/estimation-equations). Therefore, the OH exposure equations derived based on the "Penn State" lamps (Li et al., 2015;Peng et al., 2015;Peng et al., 2016) are not applicable to this OFR. The OH exposures based on calibration using 500 ppb SO2 represent an upper limit of the actual OH exposures inside the OFR. The following sentence is added to the text (Lines 163-165):

Due to external OH reactivity from CO,  $NO_x$ , and other reactants, these  $OH_{exp}$  levels represent upper limits of the actual OH exposures inside the OFR (Li et al., 2015;Peng et al., 2015).

6. In the introduction, the previous relevant researches before this work should be well reviewed.

**Response**:**

• This paper pioneers the EF comparisons between fresh and aged multipollutant mixtures through laboratory controlled burns with atmospheric aging simulated by an oxidation flow reactor. Relevant past studies for both in-situ field and laboratory peat combustion have been reviewed and cited with a total of 31 references. Where appropriate, the EFs for peat burning are compared throughout the text. Since most peat burning measurements were conducted in Southeast Asia, the following sentence and references are added in the "Introduction" section (Lines 69-73):

[revised manuscript text omitted]

|                                  |           | Emission Factors in g/kg |                   |                    |                                  |                    |                   |                    |                   |                    |                     |                    |                   |                    |                    |                    |                    | 1                    |                     |
|----------------------------------|-----------|--------------------------|-------------------|--------------------|----------------------------------|--------------------|-------------------|--------------------|-------------------|--------------------|---------------------|--------------------|-------------------|--------------------|--------------------|--------------------|--------------------|----------------------|---------------------|
|                                  |           | EF                       | M2.5 EF           |                    | F OC EF EC |                    |                   | EF TC I |                   |                    | WSOC EFLevoglucosan |                    |                   | an WSOC/OC         |                    | WSOC/PM2.5         |                    | OC/PM 2.5 |                     |
| Peat Type                        | Sample ID | FRESH a       | AGED a | FRESH a | AGED a                | FRESH a | AGED a | FRESH a | AGED a | FRESH a | AGED a   | FRESH a | AGED a | FRESH a | AGED a  | FRESH a | AGED a  | FRESH a   | AGED a   |
|                                  | PEAT030   | 44.937                   | 39.709            | 27.157             | 20.511                           | 1.057              | 0.629             | 28.214             | 21.140            | 17.753             | 13.680              | 19.703             | 12.880            | 65.37%             | 66.70%             | 39.51%             | 34.45%             | 60.43%               | 51.65%              |
|                                  | PEAT031   | 42.173                   | 33.042            | 25.645             | 16.629                           | 0.160              | 0.611             | 25.805             | 17.240            | 14.391             | 10.785              | 14.330             | 9.059             | 56.11%             | 64.86%             | 34.12%             | 32.64%             | 60.81%               | 50.33%              |
| Odintsovo, Russia                | PEAT032   | 43.117                   | 46.130            | 22.248             | 15.632                           | 1,177              | 0.521             | 23,425             | 16.153            | 16.075             | 13.064              | 12.174             | 6.922             | 72.26%             | 83.58%             | 37.28%             | 28.32%             | 51.60%               | 33.89%              |
| (n=6)                            | PEAT033   | 40.491                   | 47.894            | 26.964             | 16.901                           | 0.515              | 1.011             | 27.479             | 17.912            | 15.453             | 11.415              | 16.263             | 7.612             | 57.31%             | 67.54%             | 38.16%             | 23.83%             | 66.59%               | 35.29%              |
|                                  | PEAT034   | 50.522                   | 45.404            | 28,736             | 20.008                           | 0.822              | 0.532             | 29,558             | 20.540            | 16.962             | 12.552              | 18.328             | 8,660             | 59.03%             | 62.73%             | 33,57%             | 27.64%             | 56.88%               | 44.07%              |
|                                  | PEAT035   | 34.649                   | 31.065            | 20.019             | 13.467                           | 0.900              | 0.823             | 20.919             | 14.290            | 12.320             | 11.234              | 13.704             | 7.432             | 61.54%             | 83.42%             | 35.56%             | 36.16%             | 57.78%               | 43.35%              |
| Average ± SD                     |           | $42.65 \pm 5.22$         | $40.54 \pm 7.15$  | $25.13 \pm 3.32$   | $17.19 \pm 2.67$                 | $0.77 \pm 0.38$    | $0.69 \pm 0.19$   | $25.90 \pm 3.23$   | $17.88 \pm 2.61$  | $15.49 \pm 1.94$   | $12.12 \pm 1.15$    | $15.75 \pm 2.88$   | $8.76 \pm 2.17$   | $61.94 \pm 6.04\%$ | $71.47 \pm 9.46\%$ | 36.37 ± 2.34%      | $30.51 \pm 4.68\%$ | $59.01 \pm 4.97\%$   | $43.10 \pm 7.38\%$  |
| C.o.V.                           |           | 12.24%                   | 17.64%            | 13.22%             | 15.53%                           | 48.66%             | 27.92%            | 12.47%             | 14.58%            | 12.55%             | 9.46%               | 18.30%             | 24.77%            | 9.75%              | 13.24%             | 6.44%              | 15.35%             | 8.43%                | 17.12%              |
|                                  | PEAT023   | 39.719                   | 39.648            | 27.252             | 23.769                           | 1.200              | 1.289             | 28.452             | 25.058            | 8.901              | 9.395               | 1.615              | 1.594             | 32.66%             | 39.53%             | 22.41%             | 23.70%             | 68.61%               | 59.95%              |
|                                  | PEAT025   | 32.975                   | 27.337            | 26.845             | 16.698                           | 0.844              | 0.598             | 27.688             | 17.296            | 7.647              | 7.736               | 3.041              | 1.023             | 28.49%             | 46.33%             | 23.19%             | 28.30%             | 81.41%               | 61.08%              |
| Pskov, Siberia                   | PEAT026   | 34.739                   | 21.511            | 24.636             | 17.981                           | 1.431              | 0.824             | 26.067             | 18.805            | 9.003              | 8.094               | 2.327              | 1.048             | 36.54%             | 45.02%             | 25.91%             | 37.63%             | 70.92%               | 83.59%              |
| (n=6)                            | PEAT027   | 27.468                   | 18.934            | 21.274             | 11.310                           | 0.000              | 0.459             | 21.274             | 11.769            | 9.065              | 7.385               | 2.358              | 0.719             | 42.61%             | 65.30%             | 33.00%             | 39.00%             | 77.45%               | 59.73%              |
|                                  | PEAT028   | 42.074                   | 45.032            | 31.372             | 22.388                           | 0.069              | 0.949             | 31.441             | 23.337            | na b    | 9.580               | 3.301              | 1.555             | na b    | 42.79%             | na b    | 21.27%             | 74.57%               | 49.72%              |
|                                  | PEAT029   | 26.547                   | 31.524            | 24.397             | 16.274                           | 0.620              | 0.535             | 25.017             | 16.809            | 8.494              | 9.258               | 3.120              | 1.309             | 34.82%             | 56.89%             | 32.00%             | 29.37%             | 91.90%               | 51.62%              |
| Average ± SD                     |           | 33.92 ± 6.29             | 30.66 ± 10.20     | 25.96 ± 3.40       | $18.07 \pm 4.52$                 | $0.69 \pm 0.58$    | $0.78 \pm 0.31$   | 26.66 ± 3.44       | $18.85 \pm 4.80$  | $8.62 \pm 0.59$    | $8.57 \pm 0.95$     | $2.63 \pm 0.64$    | $1.21\pm0.34$     | 35.02 ± 5.20%      | $49.31 \pm 9.78\%$ | 27.30 ± 4.93%      | 29.88 ± 7.19%      | $77.48 \pm 8.41\%$   | $60.95 \pm 12.07\%$ |
| C.o.V.                           |           | 18.54%                   | 33.27%            | 13.11%             | 24.99%                           | 84.02%             | 40.19%            | 12.91%             | 25.49%            | 6.83%              | 11.06%              | 24.41%             | 28.15%            | 14.85%             | 19.84%             | 18.06%             | 24.06%             | 10.86%               | 19.80%              |
|                                  | PEAT013   | 17.305                   | 16.320            | 14.619             | 13.122                           | 0.543              | 0.338             | 15.162             | 13.460            | 6.877              | 5.314               | 1.790              | 2.269             | 47.04%             | 40.50%             | 39.74%             | 32.56%             | 84.48%               | 80.41%              |
| Number Alerte UCA                | PEAT014   | 30.090                   | 27.119            | 22.257             | 17.531                           | 0.640              | 0.884             | 22.897             | 18.415            | 7.294              | 7.413               | 4.297              | 4.195             | 32.77%             | 42.29%             | 24.24%             | 27.34%             | 73.97%               | 64.64%              |
| (n=5)                            | PEAT019   | 27.733                   | 29.102            | 20.459             | 18.942                           | 0.272              | 0.378             | 20.731             | 19.319            | 6.651              | 7.321               | 8.043              | 6.285             | 32.51%             | 38.65%             | 23.98%             | 25.16%             | 73.77%               | 65.09%              |
| (1-5)                            | PEAT020   | 30.406                   | 28.527            | 17.196             | 15.718                           | 0.932              | 1.090             | 18.127             | 16.808            | 7.190              | 8.167               | 2.726              | 1.971             | 41.81%             | 51.96%             | 23.65%             | 28.63%             | 56.55%               | 55.10%              |
|                                  | PEAT022   | 14.390                   | 23.086            | 12.270             | 9.053                            | 0.593              | 0.063             | 12.863             | 9.116             | 5.686              | 5.287               | 1.525              | 0.652             | 46.34%             | 58.40%             | 39.51%             | 22.90%             | 85.27%               | 39.21%              |
| Average ± SD                     |           | $23.98 \pm 7.57$         | $24.83 \pm 5.31$  | 17.36 ± 4.09       | $14.87\pm3.91$                   | $0.60 \pm 0.24$    | $0.55\pm0.42$     | $17.96 \pm 4.06$   | $15.42\pm4.17$    | $6.74 \pm 0.64$    | $6.70 \pm 1.32$     | $3.68 \pm 2.67$    | $3.07\pm2.20$     | $40.09 \pm 7.10\%$ | $46.36\pm8.47\%$   | $30.22 \pm 8.59\%$ | $27.32 \pm 3.65\%$ | $74.81 \pm 11.60\%$  | $60.89 \pm 15.13\%$ |
| C.o.V.                           |           | 31.56%                   | 21.37%            | 23.58%             | 26.31%                           | 39.58%             | 76.78%            | 22.60%             | 27.06%            | 9.52%              | 19.69%              | 72.67%             | 71.44%            | 17.70%             | 18.26%             | 28.41%             | 13.38%             | 15.50%               | 24.85%              |
|                                  | PEAT008   | 57.197                   | 54.119            | 37.217             | 39.897                           | 1.314              | 1.064             | 38.531             | 40.961            | 9.282              | 10.598              | 1.807              | 1.487             | 24.94%             | 26.56%             | 16.23%             | 19.58%             | 65.07%               | 73.72%              |
| Putnam County Lakebed, Florida   | PEAT009   | 46.012                   | 42.248            | 36.055             | 28.139                           | 2.177              | 0.923             | 38.232             | 29.062            | 10.505             | 10.919              | 1.448              | 1.185             | 29.13%             | 38.80%             | 22.83%             | 25.84%             | 78.36%               | 66.60%              |
| (n=4)                            | PEAT005   | 48.798                   | 57.969            | 34.423             | 37.209                           | 0.922              | 0.909             | 35.345             | 38.119            | 8.369              | 14.017              | 1.439              | 0.659             | 24.31%             | 37.67%             | 17.15%             | 24.18%             | 70.54%               | 64.19%              |
|                                  | PEAT006   | 60.509                   | 61.350            | 38.850             | 43.910                           | 0.898              | 0.913             | 39.748             | 44.823            | 9.380              | 13.576              | 1.991              | 1.133             | 24.14%             | 30.92%             | 15.50%             | 22.13%             | 64.20%               | 71.57%              |
| Average ± SD                     |           | $53.13\pm 6.84$          | $53.92 \pm 8.32$  | $36.64 \pm 1.87$   | $37.29 \pm 6.69$                 | $1.33\pm0.60$      | $0.95\pm0.07$     | $37.96 \pm 1.86$   | $38.24\pm6.71$    | $9.38 \pm 0.87$    | $12.28 \pm 1.77$    | $1.67\pm0.27$      | $1.12\pm0.34$     | $25.63 \pm 2.36\%$ | $33.49 \pm 5.78\%$ | $17.93 \pm 3.34\%$ | $22.93 \pm 2.70\%$ | $69.54 \pm 6.51\%$   | $69.02 \pm 4.39\%$  |
| C.o.V.                           |           | 12.88%                   | 15.44%            | 5.10%              | 17.95%                           | 44.99%             | 7.84%             | 4.91%              | 17.54%            | 9.32%              | 14.40%              | 16.38%             | 30.68%            | 9.21%              | 17.27%             | 18.62%             | 11.78%             | 9.37%                | 6.36%               |
|                                  | PEAT010   | 18.275                   | 34.139            | 17.884             | 15.235                           | 1.182              | 0.449             | 19.066             | 15.684            | 7.744              | 8.662               | 0.502              | 0.000             | 43.30%             | 56.86%             | 42.38%             | 25.37%             | 97.86%               | 44.63%              |
|                                  | PEAT011   | 28.566                   | 25.526            | 15.597             | 16.153                           | 1.260              | 0.569             | 16.857             | 16.722            | 6.547              | 6.783               | 0.452              | 0.311             | 41.98%             | 41.99%             | 22.92%             | 26.57%             | 54.60%               | 63.28%              |
| Everglades National Park Elorida | PEAT012   | 16.258                   | 15.327            | 12.117             | 9.890                            | 0.474              | 0.376             | 12.591             | 10.266            | 4.528              | 4.451               | 0.000              | 0.000             | 37.37%             | 45.01%             | 27.85%             | 29.04%             | 74.53%               | 64.53%              |
| (n=7)                            | PEAT015   | 29.133                   | 28.619            | 25.568             | 22.561                           | 1.110              | 0.670             | 26.678             | 23.230            | 9.816              | 9.891               | 0.000              | 0.635             | 38.39%             | 43.84%             | 33.69%             | 34.56%             | 87.77%               | 78.83%              |
|                                  | PEAT016   | 21.566                   | 38.113            | 19.459             | 20.460                           | 0.696              | 1.279             | 20.155             | 21.739            | 8.001              | 9.232               | 0.480              | 0.000             | 41.12%             | 45.12%             | 37.10%             | 24.22%             | 90.23%               | 53.68%              |
|                                  | PEAT017   | 24.871                   | 48.690            | 21.387             | 23.115                           | 0.752              | 0.615             | 22.139             | 23.730            | 7.153              | 9.802               | 0.389              | 0.154             | 33.44%             | 42.41%             | 28.76%             | 20.13%             | 85.99%               | 47.47%              |
|                                  | PEAT018   | 26.755                   | 43.879            | 21.147             | 23.937                           | 0.000              | 0.727             | 21.147             | 24.664            | 9.934              | 11.183              | 0.772              | 0.673             | 46.97%             | 46.72%             | 37.13%             | 25.49%             | 79.04%               | 54.55%              |
| Average ± SD                     |           | $23.63 \pm 5.05$         | 33.47 ± 11.39     | 19.02 ± 4.36       | $18.76 \pm 5.18$                 | $0.78 \pm 0.45$    | $0.67 \pm 0.30$   | $19.80 \pm 4.39$   | $19.43 \pm 5.34$  | $7.67 \pm 1.88$    | 8.57 ± 2.26         | $0.37 \pm 0.28$    | $0.25 \pm 0.30$   | $40.37 \pm 4.40\%$ | $45.99 \pm 5.06\%$ | 32.83 ± 6.69%      | $26.48 \pm 4.46\%$ | 81.43 ± 14.03%       | $58.14 \pm 11.72\%$ |
| C.o.V.                           |           | 21.38%                   | 34.02%            | 22.90%             | 27.61%                           | 57.45%             | 44.12%            | 22.19%             | 27.47%            | 24.52%             | 26.39%              | 75.66%             | 117.11%           | 10.90%             | 11.01%             | 20.36%             | 16.85%             | 17.23%               | 20.16%              |
|                                  | PEAT036   | 19.068                   | 15.123            | 15.517             | 13.240                           | 0.166              | 0.385             | 15.683             | 13.624            | 2.665              | 3.710               | 0.479              | 0.239             | 17.17%             | 28.02%             | 13.97%             | 24.53%             | 81.38%               | 87.55%              |
| Borneo, Malaysia                 | PEAT038   | 26.513                   | 23.955            | 20.370             | 14.800                           | 0.203              | 0.441             | 20.573             | 15.241            | 4.050              | 5.086               | 0.671              | 0.713             | 19.88%             | 34.37%             | 15.28%             | 21.23%             | 76.83%               | 61.78%              |
| (n=4)                            | PEAT039   | 21.895                   | 25.917            | 17.835             | 12.836                           | 0.382              | 0.346             | 18.217             | 13.181            | 4.189              | 5.794               | 0.881              | 0.621             | 23.49%             | 45.14%             | 19.13%             | 22.36%             | 81.46%               | 49.53%              |
|                                  | PEAT041   | 23.073                   | 25.276            | 18.473             | 16.566                           | 0.352              | 0.000             | 18.824             | 16.566            | 3.500              | 5.924               | 1.090              | 0.673             | 18.95%             | 35.76%             | 15.17%             | 23.44%             | 80.06%               | 65.54%              |
| Average ± SD                     |           | $22.64 \pm 3.08$         | $22.57 \pm 5.03$  | $18.05 \pm 2.00$   | $14.36 \pm 1.70$                 | $0.28 \pm 0.11$    | $0.29 \pm 0.20$   | $18.32 \pm 2.02$   | $14.65 \pm 1.55$  | $3.60 \pm 0.69$    | $5.13 \pm 1.01$     | $0.78 \pm 0.26$    | $0.56 \pm 0.22$   | 19.87 ± 2.66%      | 35.82 ± 7.06%      | 15.89 ± 2.24%      | $22.89 \pm 1.42\%$ | 79.93 ± 2.16%        | 66.10 ± 15.85%      |
| C.o.V.                           |           | 13.62%                   | 22.29%            | 11.09%             | 11.82%                           | 38.90%             | 68.01%            | 11.05%             | 10.59%            | 19.20%             | 19.78%              | 33.85%             | 38.88%            | 13.38%             | 19.72%             | 14.11%             | 6.19%              | 2.71%                | 23.98%              |
|                                  |           |                          |                   |                    |                                  |                    |                   |                    |                   |                    |                     |                    |                   |                    |                    |                    |                    |                      |                     |
| All 25 % Peat Samples:
(n=32) |           |                          |                   |                    |                                  |                    |                   |                    |                   |                    |                     |                    |                   |                    |                    |                    |                    |                      |                     |
| Average ± SD                     |           | 32.74 ± 12.07            | 34.11 ± 12.45     | 23.29 ± 6.97       | $19.50 \pm 8.09$                 | $0.74 \pm 0.49$    | $0.66 \pm 0.31$   | $24.03 \pm 7.17$   | $20.16\pm8.27$    | 8.88 ± 3.90        | $8.98 \pm 2.85$     | $4.41 \pm 5.86$    | $2.61 \pm 3.35$   | 39.09 ± 14.23%     | 48.61 ± 14.37%     | 28.09 ± 8.83%      | 27.11 ± 5.17%      | 73.78 ± 11.87%       | 58.63 ± 13.58%      |
| C.o.V.                           |           | 36.87%                   | 36.50%            | 29.92%             | 41.50%                           | 65.73%             | 47.50%            | 29.85%             | 41.01%            | 43.84%             | 31.72%              | 132.87%            | 128.31%           | 36.41%             | 29.55%             | 31.43%             | 19.06%             | 16.09%               | 23.16%              |
|                                  |           |                          |                   |                    |                                  |                    |                   |                    |                   |                    |                     |                    |                   |                    |                    |                    |                    |                      | •                   |
| Putnam County I skehed Florida   | PEAT042   | 39,744                   | 37.143            | 31.436             | 20.836                           | 1.043              | 0.434             | 32.479             | 21.270            | 8.024              | 7.610               | 1.629              | 0.823             | 25.52%             | 36.52%             | 20.19%             | 20.49%             | 79.10%               | 56.10%              |
| (60 % moisture content)          | DEAT042   | 26 704                   | 24.070            | 20.500             | 24.842                           | 0.015              | 1 927             | 20.421             | 26.660            | 9 126              | 7 979               | 1 456              | 0.024             | 27.54%             | 21 71%             | 22.14%             | 22.52%             | 90 20%               | 71.04%              |
| (n=3)                            | PEA1045   | 30.704                   | 34.970            | 29.306             | 24.842                           | 0.915              | 1.627             | 30.421             | 20.009            | 8.120              | 1.010               | 1.430              | 0.924             | 27.34%             | 51./1%             | 22.14%             | 22.33%             | 80.39%               | /1.04%              |
| 1                                | PFAT044   | 31 344                   | 31.960            | 27.063             | 20 761                           | 1.028              | 0.276             | 28 090             | 21.038            | 14 577             | 9.025               | 1.038              | 0.798             | 53 86%             | 43 47%             | 46 51%             | 28 24%             | 86 34%               | 64.96%              |

Table S7. Individual and averaged emission factors for fresh vs. aged PM2.5 mass and carbon.

 Co.v.
 District and Fresh
 Co.v.
 District and Fresh
 Co.v.
 Co.v.

Average ± SD

13.86%

36.65%

9.20%

22.14%

7.85%

44.36%

15.88%

7.24%

 $0.85 \pm 0.07 \quad 35.64 \pm 15.81\% \quad 37.23 \pm 5.91\% \quad 29.61 \pm 14.66\% \quad 23.75 \pm 4.02\% \quad 81.94 \pm 3.86\% \quad 64.03 \pm 7.51\% \quad 10.05 \pm 10.05\% \quad 10.0$

16.91%

4.72%

11.73%

49.52%

35.93 ± 4.25 34.69 ± 2.60 29.33 ± 2.19 22.15 ± 2.33 1.00 ± 0.07 0.85 ± 0.85 30.33 ± 2.20 22.99 ± 3.19 10.24 ± 3.75 8.17 ± 0.75 1.37 ± 0.30

100.90%